


# Molecular markers of biomass burning and primary biological aerosols in urban Beijing: Size distribution and seasonal variation

Shaofeng Xu[1,2], Lujie Ren[1], Yunchao Lang[1], Shengjie Hou[2], Hong Ren[1,2], Lianfang Wei[2], Libin Wu[1], Junjun Deng[1], Wei Hu[1], Xiaole Pan[2], Yele Sun[2], Zifa Wang[2], Hang Su[3], Yafang Cheng[3], and Pingqing Fu[1,2,3]

[1]Institute of Surface-Earth System Science, Tianjin University, Tianjin, 300072, China
[2]LAPC, Institute of Atmospheric Physics, Chinese Academy of Sciences, Beijing 100029, China
[3] Multiphase Chemistry Department, Max Planck Institute for Chemistry, Mainz, Germany

**Correspondence:** Yunchao Lang (yunchao.lang@tju.edu.cn) and Pingqing Fu (fupingqing@tju.edu.cn)

**Abstract.** Biomass burning and primary biological aerosol particles account for an important part of urban aerosols. Floods of studies have been conducted on the chemical compositions of fine aerosols ($PM_{2.5}$) in megacities where the haze pollution is one of the severe environmental issues in China. However, little is known about their size distributions in atmospheric aerosols in the urban boundary layer. Here, size-segregated aerosol samples were collected in Beijing during haze and clear days from April 2017 to January 2018. Three anhydrosugars, six primary saccharides and four sugar alcohols in these samples were identified and quantified by gas chromatography/mass spectrometry (GC/MS). Higher concentrations of a biomass burning tracer, levoglucosan, were detected in autumn and winter than other seasons. Sucrose, glucose, fructose, mannitol and arabitol were more abundant in the bloom and glowing seasons. Particularly high level of trehalose was found in spring, which was largely associated with the Asian dust outflows. Anhydrosugars, xylose, maltose, inositol and erythritol are mainly existed in the fine mode (<2.1 μm), while the others showed the coarse mode privilege. The concentrations of measured tracers of biomass burning particles and primary biological particles in the haze events were higher than those in the non-hazy days, with an enrichment factor of 2–10. Geometric mean diameters (GMD) of molecular markers of biomass burning and primary biological aerosols showed that there was no significant difference in the coarse mode (>2.1 μm) between the haze and non-haze samples, while a size shift towards large particles and large GMDs in the fine fraction (<2.1 μm) was detected during the hazy days, which highlights that the stable meteorological conditions with high relative humidity in urban Beijing may favor the condensation of organics onto coarse particles.The contributions of reconstructed primary organic carbon (POC) by tracer-based methods from plant debris, fungal spores, and biomass burning to aerosol OC in the total mode particles were in the ranges of 0.09–0.30% (on average 0.21%), 0.13–1.0% (0.38%) and 1.2–7.5% (4.5%), respectively. This study demonstrates that the contribution of biomass burning was significant in Beijing throughout the whole year with the predominance in the fine mode, while the contributions of plant debris and fungal spores dominated in spring and summer in the coarse mode, especially in the size of >5.8 μm. Our observations demonstrate that the sources, abundance,



and chemical composition of urban aerosol particles are strongly size dependant in Beijing, which are important to better understand their environmental and health effects of urban aerosols and should be considered in air quality and climate models.

# 1 Introduction

Organic aerosols (OAs) account for 20–90% of atmospheric aerosols in the atmosphere. Numerous studies found that OAs have significant impacts on the hydrological cycle and global climate (Jimenez et al., 2009;Ng et al., 2010;Pöschl, 2005), as well as human health (Nel, 2005). Concentration, composition and size of OAs are vital factors for such adverse impacts (Tremblay et al., 2007). Fine aerosol particles are more detrimental to human health due to the ability to deposit into deep parts of lungs (Nemmar et al., 2002). In addition, due to distinct sources of fine and coarse particles, particle size of organic species existing in could act as a tool to identify OA sources and to assess health hazard.

Primary biological aerosol particles (PBAP) and biomass burning (BB) are two major source of atmospheric primary organic aerosols (POA). PBAPs those are directly emitted into the atmosphere include bacteria, plant debris, fungal spores, fragments of plants and animals (Després et al., 2012;Fang et al., 2018;Fu et al., 2012;O'dowd et al., 2004;Samaké et al., 2019). These PBAPs not only affect the hydrological cycle and climate change by means of acting as cloud and ice nuclei but also jeopardize environmental and human health (Fröhlich-Nowoisky et al., 2012;Fröhlich-Nowoisky et al., 2016;Humbal et al., 2018;Rajput et al., 2018). Although recent studies have revealed the abundance and size distributions of PBAPs, quantitate studies regarding their emission sources and relative contributions remain poorly known (China et al., 2018;Di Filippo et al., 2013;Huffman et al., 2010;Jia et al., 2010;Tobo et al., 2013). Primary saccharides have been used as tracers for PBAP (Fu et al., 2013;Medeiros et al., 2006;Wan et al., 2007). Sucrose, glucose and fructose are predominantly derived from plant materials, such as pollen, fruit, and leaves (Pacini, 2000;Puxbaum and Tenze-Kunit, 2003). Mannitol, arabitol, and trehalose have been recognized as tracers of airborne fungi (Bauer et al., 2008;Fu et al., 2016;Samaké et al., 2019;Verma et al., 2018).

Biomass burning (BB), as a substantial source of particles in the atmosphere, generate plenty of POA (Andreae and Merlet, 2001;Wan et al., 2019;Yan et al., 2015). Levoglucosan, generated by the thermal decomposition of cellulose (Simoneit, 2002;Simoneit et al., 1999), is regarded as the most typical tracer for BB (Simoneit et al., 2004;Simoneit et al., 1999). Isomeric ratios of individual anhydrosugars have been applied to distinguish potential biofuel sources (Engling et al., 2006;Fine et al., 2004a). The levoglucosan to mannosan ratio (L/M) varies for aerosols generated by burning of hardwood, softwood and agricultural waste (Engling et al., 2006;Fu et al., 2012;Sang et al., 2013;Sheesley et al., 2003).

Beijing has been facing serious air pollution since the last decade as it is located within a highly developed economic zone in China. Though the government has made great efforts to improve the air quality in Beijing, haze events still appear frequently, especially in winter (Cheng et al., 2016;Zheng et al., 2015). Moreover, air pollution, accompanied with high abundances of fine particles, has become a serious environmental problem in China's cities and adjacent areas. There have been many studies on the composition and sources of aerosols in Beijing (Yang et al., 2016). However, few studies have been



conducted regarding the size distribution, contribution and processes of POAs in different seasons in Beijing. Thus, research on chemical compositions and size distributions of PBAP and BB in Beijing are vital in terms of regional air quality.

In this study, size-segregated aerosols were sampled in Beijing from April 2017 to January 2018. Molecular characteristics, size distributions, and seasonal trends of anhydrosugars and other saccharides were investigated. Concentrations and contributions of plant debris, fungal spores and biomass burning derived OC were estimated according to the tracer methods. The variations in size distributions of organic species in aerosols in different seasons were also assessed.

## 2 Experimental methods

### 2.1. Sample collection

All samples were collected using a nine-stage cascade impactor sampler (Andersen, U.S.A.) at a flow rate of 25.8 L/min from April 2017 to January 2018. The sampler was put on the rooftop of a second-story building in the Institute of Atmospheric Physics, Chinese Academy of Sciences in Beijing, China (39°58'28''N, 116°22'16''E). Before sampling, the quartz fiber filters were combusted at 450°C for 6 h. The cutoff sizes are in the following sequence: >9.0, 9.0–5.8, 5.8–4.7, 4.7–3.3, 3.3–2.1, 2.1–1.1, 1.1–0.7, 0.7–0.4, and <0.4 μm. Four sets of size-segregated aerosol samples per season and thus sixteen sets (i.e., 144 samples) were obtained throughout the year. Blank filters were obtained for each season for quality control. Specific information including weather condition, sampling periods and sample numbers was summarized in Table S1. The aerosol samples were stored at a –20°C in dark before analysis.

### 2.2. Sample extraction and derivatization

Using a disinfected scissor, each filter aliquot (~6.4 cm$^2$) was cut and extracted three times with dichloromethane/methanol mixture (2:1, v/v) under ultrasonic agitation for 10 minutes. After filtering, the solvent extracts were concentrated with a rotary evaporator and a stream of pure nitrogen gas. The extracts were converted to their trimethylsilyl derivatives using a mixture of 25 μL of N,O-bis-(trimethylsilyl) trifluoroacetamide (BSTFA) with 1% trimethylsilyl chloride and 5 μL of pyridine for 3 h at 70°C (Fu et al., 2008). The derivatives were diluted by adding 30 μL internal standard (1.43 ng μL$^{-1}$ of C$_{13}$ $n$-alkane), and then were stored in a freezer before GC/MS analysis.

### 2.3. Gas chromatography/mass spectrometry measurements

In terms of GC/MS analysis, an Agilent model 7890 GC coupled to 5975C mass-selective detector was used to analyze organic compounds of these samples. Two μL of each sample was injected and separated in a fused silica capillary column. The column temperature program was optimized with a serial of procedures: start and hold for 2 min at 50 °C, increase to 120 °C with a pace at 15 °C min$^{-1}$, then raise to 300 °C at 5 °C min$^{-1}$, and keep isothermal at 300 °C for 16 min. The mass



spectrometer was operated on the Electron Ionization (EI) mode at 70 eV and scanned from 50 to 650 Dalton. Individual compound was identified by authentic standards. All the reported concentrations were corrected for the field blanks.

## 2.4. Measurements of OC and EC

The measurements of organic carbon (OC) and elemental carbon (EC) were performed using a thermal/optical carbon analyzer (model RT-4, Sunset Laboratory Inc., USA). A small punch (17-mm diameter) of each sample/blank filter was used for the determination. The concentrations of OC and EC here were all corrected for field blanks.

## 2.5. Backward trajectories and fire counts

To better characterize the features and origins of air masses in Beijing, 3-day backward trajectory analyses were conducted using the HYSPLIT4 model (Draxler and Rolph, 2013) for all the dust storm and haze periods in our study. Backward trajectories were calculated at an altitude of 500 m above ground level. Influence of BB in urban Beijing are demonstrated by fire spots acquired from the MODIS website (https://earthdata.nasa.gov/earth-observation-data/near-real-time/firms). The crowding level of the fire spots here represented the scales and intensity of BB. Cluster analyses were applied for typical dust storm and haze events in the present study (Fig. S1).

# 3 Results and discussion

## 3.1. Abundance and seasonal trends of anhydrosugars

### 3.1.1 Anhydrosugars

Levoglucosan is formed from the thermal decomposition of cellulose during vegetation burning activities, which has been used as a typical biomass burning tracer in many previous studies conducted in urban and rural locations (Cheng et al., 2013;Chowdhury et al., 2007;Fu et al., 2010;Wang et al., 2006;Yttri et al., 2007). As another important type of anhydrosugars, isomers of levoglucosan, including mannosan and galactosan, are exclusively derived from the pyrolysis of hemicellulose (Fraser et al., 2000; Simoneit, 2002). Concentrations of and temporal variations in anhydrosugars (i.e., Levoglucosan, mannosan and galactosan) are shown in Table 1 and Fig. 1. Among the three anhydrosugars, levoglucosan was the predominant compounds (80.1–1230 ng m$^{-3}$, 365 ng m$^{-3}$), which was one order of magnitude higher than mannosan (8.41–168 ng m$^{-3}$, 46.8 ng m$^{-3}$) and galactosan (5.09–108 ng m$^{-3}$, 27.6 ng m$^{-3}$). The concentrations of levoglucosan was the highest in winter, followed by autumn, spring and summer. Similar seasonality has been documented in several studies in urban Beijing (Chen et al., 2013;Liang et al., 2016). The abundances in summer and winter was comparable with previous studies (Cheng et al., 2013;Zhang et al., 2008). Mannosan and galactosan displayed a similar seasonal trend. Resemble to levoglucosan, total anhydrosugars peaked in winter (250–1503 ng m$^{-3}$, 678 ng m$^{-3}$) and bottomed in summer (87.8–246 ng m$^{-3}$, 137 ng m$^{-3}$). The



high abundance of anhydrosugars in winter is possibly due to the enhanced sources from field biomass burning and domestic heating activities, together with the frequent development of inversion layers in winter. Interestingly, the concentrations of levoglucosan in autumn (224–759 ng m$^{-3}$, 501 ng m$^{-3}$) and winter (131–1230 ng m$^{-3}$, 559 ng m$^{-3}$) were much higher than those in summer (80.1–203 ng m$^{-3}$, 114 ng m$^{-3}$). Except for the decrease of biomass burning activities, a higher level of OH radical in summer on account of high temperature and intensive ultraviolet light (Stone et al., 2012) may lead to the photochemical degradation of levoglucosan (Arangio et al., 2015;Hennigan et al., 2010;Lai et al., 2014;Mochida et al., 2010).

Concentrations of individual anhydrosugars in haze days were much higher than those in non-haze days during each season (Table 2). Moreover, the level of anhydrosugars (2310 ng m$^{-3}$, average) in winter was one order of magnitude higher than other seasons (302 ng m$^{-3}$, 171 ng m$^{-3}$, 798 ng m$^{-3}$ for spring, summer and autumn, respectively). Such trends indicated that the haze samples were largely influenced by biomass burning emissions in Beijing, consistent with previous studies in Beijing (Zhang et al., 2017) and other regions such as New Delhi (India), Tasmania (Australia) and Kathmandu (Nepal) (Fu et al., 2010;Reisen et al., 2013;Wan et al., 2019).

During the dust storm periods in spring, air masses primarily originated from the middle-to-north Asia (MNA) (Fig. S1a–1b). Specifically, about 70% of air masses originated from the west-to-central Russia and passed through Mongolia, and finally arrived in Beijing. Such phenomenon likely attributed to the influence of the Asian outflow. Compared to April 17 - 19, fewer fire spots were found in the movement route of air mass in May 4 - 5. In summer, the influence of air masses from central China was strengthened. In the meantime, frequent straw burning events were found in these regions due to the harvest season (Fig. S1c–1d). However, this emission had an insignificant impact on the anhydrosugars considering the relatively low concentrations. In winter haze periods, 53–100% of the air masses originated from the north China, and the rest fraction originated from Mongolia. Interestingly, much fewer fire spots could be observed (Fig. S1g–1h) than those in summer and autumn. In winter, domestic burning and small-scale open-field burning are much intensive than other seasons. However, these two burning activities are hardly detectable by satellite, which could account for the much higher abundances of anhydrosugars.

Pearson correlations among anhydrosugars and OC, EC were also studied (Table S2). The concentrations of levoglucosan correlated positively with those of mannosan and galactosan (r > 0.98, $p < 0.001$), which suggest that the anhydrosugars were emitted from similar sources. EC is considered to be closely related with BB (Akagi et al., 2011). Positive correlations between EC and levoglucosan (r = 0.45, $p < 0.001$), mannosan (r = 0.43, $p < 0.001$) and galactosan (r = 0.43, $p < 0.001$) indicate that BB was a major source of EC in urban Beijing.

### 3.1.2 Ratios of L/M, M/G, L/OC, and L/EC

Previous surveys found that levoglucosan/mannosan (L/M) ratios vary with biofuel burning sources, those from hardwood, softwood and crop residues are often in the ranges of 13–27, 2.5–5.8 and 25–55.7, respectively (Engling et al., 2006;Fine et al., 2004;Krůmal et al., 2010;Sheesley et al., 2003). The L/M values are exhibited in Table1 and Fig. 2. The annual L/M ratios varied within a narrow range over the sampling periods (7.62–8.64, 8.27), indicating a mixture contribution of softwood and hardwood burning. The L/M ratios were relatively high in winter (7.30–11.8, average 8.64) and spring (6.34–11.8, 8.49), being





consistent with previous studies (Zhu et al., 2015a). This was probably associated with open-field biomass burning, e.g. agricultural residues and deciduous leaves. The L/M ratios were smaller in summer and autumn, along with a higher level of M/G ratios than those in winter and spring. These results were likely attributed to the higher proportion of softwood burnings and the degradation of levoglucosan (Arangio et al., 2015;Lai et al., 2014;Mochida et al., 2010).

5      The ratio of mannosan to galactosan (M/G) is used as an auxiliary means to distinguish the burning substrates because galactosan is more abundant than mannosan in smoke aerosols from crop straws, grasses, and briquettes (Fabbri et al., 2009;Oros et al., 2006;Vicente et al., 2018). The M/G ratios during all the periods were in a range of 1.59–1.88 with an average of 1.70 (Fig. 2b). The M/G ratios maximized in autumn (1.68–1.97, 1.88) and minimized in summer (1.35–1.83, 1.59). The lower ratios in summer haze days (June 30 to July 2) were in agreement with the former studies during summer BB periods 10 (Engling et al., 2006). The higher ratios in typical summer time in July (July 14-16 and July 21-23) were consistent with previous results in North China Plain and Western United States (Fine et al., 2004b;Fu et al., 2008). The M/G ratios ranged from 1.49 to 1.89 (average 1.62) in winter, consistent with the higher abundance of galactosan over mannosan associated with the BB products from crop wastes in the North China Plain (Fu et al., 2008).

     The size distributions of L/M, M/G, L/OC, and L/EC ratios are presented in Fig. 3. Higher L/M values were observed in 15 fine particles during non-haze days, which were 1.02–1.71 times higher than those in coarse modes. In contrast, the L/M ratios did not vary significantly according to particle size in the haze days. The L/M values in fine modes during haze periods were slightly higher than those in coarse modes. The ratios between fine and coarse modes were close to unity. The elevated L/M ratios in fine modes suggested that hardwood was potentially the burning substrates, while the relatively lower L/M ratios in coarse modes indicated the mixture impact of hardwood and softwood burning. The M/G ratios were also higher in fine modes than coarse modes with a factor of 1.0–2.5, suggesting large impact of biomass burning from crop straws and grasses in fine 20 modes. Moreover, such activities were more pronounced in haze days than non-haze days, implying the increasing contributions from crop straws burning in haze days.

     The ratios of levoglucosan to OC (L/OC) and EC (L/EC) are applied to evaluate the impact of BB to airborne particles and the possible photochemical aging of levoglucosan (Feng et al., 2007;Mkoma et al., 2013;Zhang et al., 2008). The ratios of 25 L/OC showed a clear temporal trend (annual mean $3.67\times10^{-3}$, Fig. 2c) with the highest values in winter ($2.79\text{–}9.15\times10^{-3}$), followed by autumn ($3.46\text{–}5.21\times10^{-3}$), spring ($0.69\text{–}6.89\times10^{-3}$) and summer ($0.73\text{–}1.14\times10^{-3}$). Similar seasonal trends have been reported in previous studies (Feng et al., 2007;Ho et al., 2014). In comparison to annual mean L/OC ratios in Guangzhou and Zhaoqing in the PRD region ($10.8\times10^{-3}$ and $27.5\times10^{-3}$, respectively) (Ho et al., 2014), the lower values of L/OC in Beijing were potentially associated with the less extensive burning activities. EC is highly related to the activities of coal combustion, 30 vehicle exhaust and biomass burning (Akagi et al., 2011;Lu et al., 2011;Waked et al., 2014). The L/EC ratios showed a similar variation pattern to the L/OC ratios. The L/EC ratios exhibited maxima in autumn ($45\text{–}107\times10^{-3}$, $83\times10^{-3}$), followed by winter ($34\text{–}118\times10^{-3}$, $73\times10^{-3}$), spring ($20\text{–}53\times10^{-3}$, $36\times10^{-3}$) and summer ($12\text{–}25\times10^{-3}$, $17\times10^{-3}$), indicating more contribution of BB to aerosols in cold seasons.



The L/OC ratios showed a uniform dependence on particle size of much higher in fine modes during all sampling periods. The average ratios of L/OC (fine modes) to L/OC (coarse modes) for four seasons were 6.31, 4.49, 11.3 and 9.47, respectively. Such a pattern was consistent with the finding that levoglucosan mostly exists in fine modes (Simoneit and Elias, 2001). The L/OC ratios in summer were a magnitude lower than other seasons that likely attributed to the less biomass burning emissions in Beijing urban areas.

The L/EC ratios also exhibited a clear pattern with high values in the fine mode and low in the coarse mode, similar to the L/OC ratios. Cao et al (2005) found that the EC in $PM_{10}$ was more abundant than in $PM_{2.5}$, while most levoglucosan was in the fine mode, resulting in low L/EC ratios in the coarse mode. Nevertheless, a high L /EC ratio was observed in the coarse mode during 4–5 May when Beijing encountered the most severe dust storm in that spring. Unlike the other samples, levoglucosan accounted for an unexpectedly high proportion in the coarse mode (discussed in 3.3.1). Meanwhile, the concentrations ($23$–$57 \times 10^{-3}$ µg m$^{-3}$) of EC in the coarse mode were much lower in the dust period than in non-haze periods ($30$–$211 \times 10^{-3}$ µg m$^{-3}$) during spring. As a consequence, low level of EC and high level of levoglucosan lead to the high L/EC ratio in the coarse mode.

## 3.2. Abundances and seasonal variations in primary saccharides and sugar alcohols

### 3.2.1 Primary saccharides

The concentrations of primary saccharides and sugar alcohols in four seasons are exhibited in Table 1 and Fig. 4. The concentrations of primary saccharides and sugar alcohols averaged 620 ng m$^{-3}$ and 139 ng m$^{-3}$, respectively. Except maltose and xylose, other primary saccharides exhibited higher abundance in spring and summer compared to autumn and winter. Among these compounds, sucrose (30.8–1590 ng m$^{-3}$, 267 ng m$^{-3}$) was the most abundant one, followed by glucose (38.9–502 ng m$^{-3}$, 162 ng m$^{-3}$) and fructose (7.97–72.5 ng m$^{-3}$, 30.1 ng m$^{-3}$).

Fructose and glucose are mainly derived from plant fragments (e.g. bark, leaves), pollen, lichen and soil dust (Dahlman et al., 2003;Fu et al., 2012;Pacini, 2000;Vesprini et al., 2002). Glucose maximized in spring (107–502 ng m$^{-3}$, 255 ng m$^{-3}$), followed by summer (184–279 ng m$^{-3}$, 234 ng m$^{-3}$), winter (60.0–139 ng m$^{-3}$, 86.4 ng m$^{-3}$) and autumn (38.9–93.3 ng m$^{-3}$, 71.2 ng m$^{-3}$). Fructose varied with a similar trend. High concentrations in spring is likely coincided with agricultural tilling activities (Simoneit et al., 2004) and comparably active vegetation in warmer season. The significant correlation between glucose and fructose (r = 0.81, $p < 0.001$) further indicated that these two compounds originated from common sources (Table S3), in consistence with the results of previous studies in East Asian areas (Chen et al., 2013;Fu et al., 2012;Zhu et al., 2015). Studies found good correlations between fructose, glucose with OC in biomass burning aerosols over Amazonia pasture site (Graham, 2002), different with this study (Table S3), which suggested that biomass burning made minor contribution to glucose and fructose in Beijing.

Sucrose, a major constituent of pollen, peaked in the spring blossom season, being coincide well with previous studies (Fu et al., 2012;Jia et al., 2010;Pacini, 2000). Like glucose and fructose, sucrose mainly emit into the air along with the



increased biological activities (Rogge et al., 2007). Accordingly, sucrose was much more predominant in spring (266–1594 ng m$^{-3}$, 743 ng m$^{-3}$), then sharply declined in winter (87.7–209 m$^{-3}$, 137 ng m$^{-3}$) and summer (30.8–275 ng m$^{-3}$, 114 ng m$^{-3}$). Interestingly, sucrose minimized in autumn (32.6–115 m$^{-3}$, 73.3 ng m$^{-3}$) rather than in summer. This likely attributed to the agricultural activities in summer. In June and July, tilling activities make crop (e.g., wheat) roots expose to the air frequently, conducing to the release of sucrose from the roots into the atmosphere.

Trehalose is present in various microorganisms (e.g., fungi and bacteria), some higher plants and invertebrates (Medeiros et al., 2006). In this study, trehalose exhibited a high abundance in spring (51.8–559 ng m$^{-3}$, 199 ng m$^{-3}$) and peaked in summer (101–378 ng m$^{-3}$, 232 ng m$^{-3}$), sharply reduced by nearly 75% in autumn and winter. Trehalose is a useful tracer for soil input to the atmosphere in regions related to agricultural activities and dust storm (He et al., 2001). Thus, the concentration of trehalose was considerably high in a dust period during 4-5 May (559 ng m$^{-3}$) and haze days during 30 June to 2 July and 12-14 July (Fig. 4h).

Xylose and maltose exhibited a totally different pattern from other primary saccharides. Xylose showed high levels in autumn (10.2–29.7 ng m$^{-3}$, 22.2 ng m$^{-3}$) and winter (11.0–32.7 ng m$^{-3}$, 20.7 ng m$^{-3}$). Xylose exists in most woods and straw as a major constituent of hemicellulose (Chen et al., 2013;Sullivan et al., 2011). Biomass burning activities in Beijing likely caused the elevated abundance of xylose, given the positive correlation between xylose and levoglucosan (r = 0.86, $p$ < 0.001). Maltose, a microbial degradation product of starch, showed a similar seasonal pattern with xylose (Medeiros et al., 2006) and positively correlated with levoglucosan (r = 0.89, $p$ < 0.001). These results suggest that biomass burning was a major source of xylose and maltose.

### 3.2.2 Sugar alcohols

The sum of sugar alcohols (29.8–418 ng m$^{-3}$, 139 ng m$^{-3}$) was much lower than primary saccharides and anhydrosugars (Table 1). In the detected compounds, mannitol showed the highest level (3.64–259 ng m$^{-3}$, 58.4 ng m$^{-3}$), followed by arabitol (5.35–149 ng m$^{-3}$, 36.7 ng m$^{-3}$), inositol (5.05–69.2 ng m$^{-3}$, 21.8 ng m$^{-3}$) and erythritol (3.16–44.1 ng m$^{-3}$, 21.8 ng m$^{-3}$). Both mannitol and arabitol exhibited maxima in summer (65.1–259 ng m$^{-3}$,151 ng m$^{-3}$ and 43.4–149 ng m$^{-3}$, 79.6 ng m$^{-3}$, respectively). Mannitol minimized (3.64–37.7 ng m$^{-3}$, 20.4 ng m$^{-3}$) in autumn and arabitol minimized in winter (5.52–33.8 ng m$^{-3}$, 14.1 ng m$^{-3}$). Mannitol significantly correlated with arabitol during the whole sample period (r = 0.83, $p$ < 0.001), indicating that they are generally related to common sources. Mannitol and arabitol primarily derive from airborne fungal spores (Fu et al., 2010;Medeiros et al., 2006;Zhang et al., 2010) and detritus of mature leaves (Pashynska et al., 2002). In summertime, fungal spores can be derived from multiple sources (e.g., plants, vegetation decomposition and agricultural activities); while in autumn and winter, fungal spores are mainly suspended from exposed surfaces (Liang et al., 2013). Moreover, a previous study point out higher leaf area index and atmospheric water vapor would facilitate the emission of mannitol (Heald and Spracklen, 2009). Together with that study, mannitol and arabitol in our study may be closely related to the activities of the terrestrial biosphere.





Erythritol and inositol was less abundant of the calculated sugar alcohols. The abundance of erythritol was the lowest in spring (3.16–15.8 ng m$^{-3}$, 9.92 ng m$^{-3}$) and the highest in summer (13.8–44.1 ng m$^{-3}$, 30.2 ng m$^{-3}$), and maintained at a relative high level in autumn and winter (11.1–33.0 ng m$^{-3}$, 22.3 ng m$^{-3}$ and 10.7–41.6 ng m$^{-3}$, 24.9 ng m$^{-3}$, respectively). Temporally, erythritol correlated well with levoglucosan (r = 0.50, $p < 0.001$), indicating a potential contribution from BB (Table S3).

Results on inositol in aerosols was seldom reported (Chen et al., 2013). In this study, inositol showed maxima (20.7–69.2 ng m$^{-3}$, 41.5 ng m$^{-3}$) in spring and minima (5.05–26.0 ng m$^{-3}$, 12.4 ng m$^{-3}$) in winter. Inositol was positively correlated with trehalose (r = 0.65, $p < 0.001$) and exhibited a similar temporal variation with other sugar alcohols, suggesting developing leaves and plant debris may be a potential source.

### 3.3. Size distribution

### 3.3.1 Anhydrosugars

The size distributions of anhydrosugars for each sampling period are exhibited in Fig. 5. The concentrations of anhydrosugars in fine (< 2.1μm) and coarse (≥ 2.1μm) mode and whole size ranges are listed in Table 3. Among the class of anhydrosugars, levoglucosan was approximately an order of magnitude higher than other compounds in the fine and coarse particles. Levoglucosan is initially emitted from the flame, subsequently condenses onto preexisting particles as temperature decrease, and thus exists mostly in fine particles (Simoneit and Elias, 2001). In this study, levoglucosan showed a unimodal

size distribution with a peak at the ranges of 0.4–2.1 μm during most sampling periods (Figs. 5a–5d). The concentrations of levoglucosan in the fine mode were 1.8–9.3 times of those in the coarse mode. Mannosan and galactosan also showed a size distribution pattern similar to levoglucosan during all sampling periods.

Interestingly, all three anhydrosugars exhibited a unimodal size distribution in the coarse mode during the most serious

dust storm period (4 to 5 May). Such distribution behaviors highlighted the significant contribution of dust storm to the loading of OA in Beijing, which can change the size distribution of organic compounds and lead to severe air pollution. Similar result reported that there is a remarkable size shift of OA from the fine mode to the coarse mode in heavily polluted days (Herner et al., 2006). The total mode GMDs (geometric mean diameters) of these species, corresponding to the dust storm periods in spring, showed larger value (1.12–1.47 μm) than those in the non-haze periods (0.62–1.04 μm). The typical summer haze

period (June 30 to July 2) with a minor peak at 3.3–5.8 μm in the coarse mode was likely due to the meteorological conditions with higher humidity and lower wind speed. During the haze period, the concentrations of all species was comparatively higher, especially in autumn and winter. As mentioned above, large scales of BB activities in cold seasons lead to massive emissions of these species. Moreover, autumn/winter were characterized by low mixing layer and stagnant conditions in haze days, which benefit the accumulation of the pollutants (Balducci and Cecinato, 2010;Bigi and Ghermandi, 2011;Carbone et al.,

2010;Perrone et al., 2012). The total and coarse mode GMDs of anhydrosugars in spring and summer were larger than in autumn and winter, while the fine mode GMDs showed smaller values (Table 4, S4 and S5). Such altering of particle size has been observed in other places (Herner et al., 2006;Wang et al., 2011).



### 3.3.2 Primary saccharides

The size distributions of all primary sugars are exhibited in Figs. 6–7. Both glucose and fructose existed mostly in coarse modes. Though the concentrations of glucose were much higher than fructose, these two compounds showed similar trends with a bimodal size distribution, peaking at 0.7–1.1 and >5.8 μm for spring, 0.4–1.1 and 3.3–5.8 μm for summer and autumn, 0.4–1.1 and 4.7–9 μm for winter. Compared to the smaller contribution in the fine mode in spring and summer (5.9–8.5%, 14.5–26.7% for glucose and fructose, respectively), the fractions of these two compounds were more abundant in fine aerosols during autumn and winter (15.0–24.4%, 32.6–45.8% for glucose and fructose, respectively). However, the origins of glucose and related sugars in fine particles remained unclear. Former studies suggested that fragmented pollen grains, either in cytoplasmic vesicles or dissolved in the cytosol, could be sources of sugars in the fine aerosol (Pacini, 2000;Yttri et al., 2007).

The size distribution of sucrose was exhibited in Figs. 6i–6l. Sucrose showed a bimodal size distribution with high concentrations in the coarse mode during the sampling periods. A minor peak was found in the size ranges of 0.4–1.1 μm and a major peak was in >5.8 μm. Compared to other three seasons (approximate 14.9%, 18.8%, 19.1% for summer, autumn and winter, respectively), sucrose existed much less in the fine mode (3.3%) in spring. Such phenomenon could be ascribed to the prevailing of pollen grain emission during the spring bloom season (Xu et al., 2012). Previous studies reported that most pollens of grasses actually remain in the anthers, and the cycle of wetting/drying cause the pollens to rupture from the anthers, with the pollen in the size ranges of 120 nm to 4.67 μm (Taylor et al., 2002). Meanwhile, furious windblown coarse aerosols from resuspension of dust in spring could further lead to higher abundance. Glucose, fructose and sucrose exhibited the largest total and coarse mode GMDs in dust storm aerosols, suggesting the effect of dust storms on particles which could contribute a large number of sugars (Table 4, Table S5).

Trehalose resembles more closely arabitol and mannitol, which existed mostly in the coarse mode with a proportion of over 90% and showed a nearly unimodal size distribution during the whole sampling periods (Figs. 7a–7d). This result indicates that trehalose originated from fungal spores, being in agreement with former studies (Lewis and Smith, 1967;Medeiros et al., 2006). The same as sucrose, trehalose was greatly influenced by the dust storm in spring. The concentrations of trehalose in dust storm were 4-10 times higher than in non-haze periods (Fig. 7a), indicating the essential impact of dust storm. Also, trehalose showed the largest total, fine and coarse GMDs in dust storm periods (Table 4, Table S4-5). This phenomenon was in good agreement with the observation in Seoul (Jeon et al., 2013). Noticeably, the total and coarse mode GMDs of trehalose in winter non-haze days were much larger than those in haze days. We assumed that soil resuspension by wind input substantial large particles into the air, while the mechanism needs to be further discussed.

Differ from other sugar and sugar alcohols, xylose and maltose showed a bimodal size distribution (Fig. 7). Except for summer, xylose and maltose were observed with a major peak at 0.4–1.1 μm and a minor peak at 3.3–5.8 μm in another three seasons. In summer, xylose and maltose existed more in coarse aerosols and even peaked at 3.3–9 μm during 30 June to 2 July. The distribution patterns of xylose and maltose were similar to those of anhydrosugars, suggesting biomass burning may be a dominant sugar source in the fine mode. Xylose and glucose were positively correlated well with each other (r = 0.83, $p$ <




0.001) in coarse modes. Whereas, the origin of maltose in coarse modes remained unclear in this study. The correlations between maltose and glucose and sucrose were poor (r =-0.04 and r=0.01, respectively), which was contrary to the former studies that maltose is particularly important in developing flower buds (Bieleski, 1995), indicating that maltose may be derived from other sources.

### 3.3.3 Sugar alcohols

The size distributions of sugar alcohols were illustrated in Figs. 6–7. Mannitol and arabitol showed similar trends while erythritol and inositol exhibited a different distribution pattern. Both mannitol and arabitol showed a unimodal size distribution in spring and summer while bimodal size distributions in autumn and winter. Mannitol and arabitol peaked at >9.0 μm and 3.3–9.0 μm in spring and summer, respectively, while they exhibited more in fine aerosols in winter. Such a phenomenon was attributed to various sources of mannitol and arabitol. In summer and autumn, fungal spores in the atmosphere can be derived from complex sources (e.g., plant leaves, agricultural activities), which may contribute more to coarse particles than those in winter (Yttri et al., 2007). While in winter, there are less plant cover and agricultural activities, resulting in a more fraction in the fine mode. Former studies suggested that mannitol was present in aerosols sampled influenced by wildfire smoke, and mannitol and arabitol might be emitted by thermal stripping during wildfires (Medeiros and Simoneit, 2007;Simoneit et al., 2004). These sources can also effectively elevate the abundance of mannitol and arabitol in the fine mode in winter. Compared to other sampling periods, mannitol and arabitol in the dust storm exhibited the largest GMD values, which was ascribed to large emissions from the re-suspension of fungal spores from the surface soil. In summer, autumn and winter haze days, the total and coarse mode GMDs of mannitol and arabitol were generally larger than those in non-haze (Table 4 and S4).

Inositol showed a bimodal size distribution in four seasons, while the size distribution pattern varied with the seasons. The concentration of inositol in the coarse mode was higher with a peak at >9.0 μm in spring. While in other seasons, the concentration of inositol peaked at 0.4–1.1 μm. The total mode GMDs of inositol in the dust storm samples were much larger than those in other non-haze and haze samples, indicating the substantial influence of dust. The size pattern of inositol was similar to those of arabitol and mannitol, potentially responding to fungal spores in local terrestrial biosphere. In this study, inositol was positively related to levoglucosan in the fine mode (r = 0.77, $p < 0.001$), which could explain the fine mode preference in cold seasons. Such a correlation was further ascertained by the total and fine mode GMDs of inositol, for its similarity with that of anhydrosugars.

Erythritol showed a bimodal size distribution for all sampling periods. In summer, a major peak was found at 3.3–5.8 μm and a minor peak at 0.4–1.1 μm. While in other seasons, the distribution patterns reversed with the prevailing of the fine mode and peaked at 0.4–1.1 μm. One explanation is that high relative humidity in summer facilitate the fine particles migrate to large particles. The source of erythritol in the fine mode was different from that in the coarse mode. The positive relationship between erythritol and levoglucosan (r = 0.90, $p < 0.001$) in the fine mode, suggests the dominant contribution from biomass burning to erythritol. However, erythritol co-varied with arabitol (r = 0.59, $p < 0.001$) in the coarse mode, suggesting that erythritol is partly of local origin.



### 3.4. Abundance and contributions of OC from biomass burning, plant debris and fungal spores

Based on the tracer methods, the quantitative estimates of OC from biomass burning (BB), plant debris and fungal spores are shown in Table 5 and Fig. 8. In a combustion test in fireplaces conducted by Fine et al. (2004b), the emission ratio of OC/levoglucosan mass was 7.35 on average, and high ratios of 8.0–8.2% are probably typical for open fires of burning grasses, woods and agricultural residues (Zhang et al., 2007). Considering the several burning of agricultural wastes, an L/OC ratio of 8.2% was applied in this study. Except for summer, BB-OC was the most dominant and showed an order of magnitude higher than the other two OC contributors. The level of BB-OC maximized in cold seasons (6.81 μg m$^{-3}$ for winter, 6.11 μg m$^{-3}$ for autumn) and minimized in summer (1.39 μg m$^{-3}$). The contributions of BB-OC reached the highest level (7.52%) in winter, followed by autumn (5.51%) and the lowest (1.21%) in summer (Table 5 and Fig. 8b). Apart from the elevated abundance of levoglucosan, stable meteorological conditions wound also lead to the accumulation of BB-OC in cold seasons (He et al., 2001).

Compared to BB-OC, the contributions of plant debris and fungal spores exhibited totally different patterns. Based on the relationship between glucose and plant debris (Puxbaum and Tenze-Kunit, 2003), the plant debris-derived OC was estimated. The plant debris-derived OC showed a seasonal trend with higher concentrations in spring (0.34 μg m$^{-3}$) and summer (0.31 μg m$^{-3}$), but lower in autumn (0.09 μg m$^{-3}$) and winter (0.11 μg m$^{-3}$). Likewise, the contributions of plant debris to OC were relative larger in spring and summer (both 0.30%), but it only accounted for 0.01% in autumn and 0.16% in winter. Mannitol, a tracer for fungal spores, was applied to estimate the contribution of fungal spores to OC using an experimentally derived factors of 1.7 pg mannitol and 13 pg OC per fungal spore (Bauer et al., 2008;Carvalho et al., 2003;Graham et al., 2003).In general, the abundance of fungal spores-derived OC exhibited a similar trend with plant debris derived-OC, which maximized in summer (1.15 μg m$^{-3}$) and reduced in autumn and winter (0.16 μg m$^{-3}$). The contributions of fungal spores to OC maximized in summer (1.00%) and minimized in autumn (0.13%).

The contributions of OC from BB, plant debris and fungal spores according to particle size are shown in Fig. 10. BB-OC accounted for more than 85% of the three species in the fine modes. It was more pronounced in autumn and winter as approximately 100%. The contributions of BB-OC in coarse modes were of two patterns. In the size range of 2.1–9.0 μm, the relative contributions of OC were higher in spring, autumn and winter with a proportion over 60%, and exhibited smallest fraction in summer. While in the size range larger than 9.0 μm, the smallest fraction of BB-OC appeared in spring and the relative contribution were below 60% in most periods. Plant debris-derived OC mostly existed in the coarse modes. In the size range larger than 9.0 μm, the plant debris-derived OC was the dominant one and peaked in spring (Fig. 9). Resemble plant debris-derived OC, fungal spores-derived OC also showed high contribution in the coarse mode. The annual contributions of fungal spores to OC ranged from 0.04–2.7% (0.56%) in the coarse mode, while those in the fine mode were just 0.03–0.14 %, 0.06%. Burege (2002) found that fungal spores in the atmosphere exist predominantly in the size range of 2–10 μm. Previous studies also found similar results that mannitol mainly occurs in the coarse size fraction (Carvalho et al., 2003;Graham et al., 2003;Samaké et al., 2019). The relative contribution decreased in the smaller size ranges and presented summer privilege over


other seasons. The contributions of fungal spores to OC were particularly high (1.48%) in the coarse fraction in summer, indicating the multiple sources of fungal spores in summertime. Both plant debris OC and fungal spores OC were present in fine modes, especially in spring and summer, which has been observed in several sites (Carvalho et al., 2003;Graham et al., 2003;Yang et al., 2012). It was worth mentioning that though the contribution of plant debris and fungal spores in the whole range were insignificant in cold seasons, their contribution in the coarse mode could not be ignored.

Compared to the contributions of BB, plant debris and fungal spores in non-haze day, the total contribution to OC showed a larger fraction in haze days for each season with enriched factors of 1.2–9.1. However, the reasons for elevated contribution in hazy days varied with season. In summer, the enhanced concentration of fungal spores-derived OC was the primary causation (Fig. 9), especially in the size range of 3.3–9.0 μm. The levels of plant debris-derived OC in haze days were about 2.4–4.0 times higher than that in non-haze days, while BB-OC and fungal spores-derived OC showed no significant differences (Figs. 8–9). Hence, the contribution to POC in haze days existed more in the coarse mode than that in non-haze days. Whereas in autumn and winter, the elevated total contribution attributed to the increased concentration of BB-OC. Fig. 9f–i showed that BB-OC in the size range of 0.4–2.1 μm of haze samples were much higher than that in non-haze samples. As a result, the prevalence of BB-OC lead to more total contribution in the fine mode and more in haze days.

## 4 Conclusions

Atmospheric concentrations, seasonal variations and size distributions of anhydrosugars and sugar alcohols were investigated in urban aerosols from Beijing, China. High concentrations of anhydrosugars in the cold seasons were associated with the enhancement of biomass burning activities and adverse meteorological conditions. High levels of primary bioaerosol tracers such as sucrose, fructose, glucose, mannitol and arabitol were found in bloom and glowing seasons. The predominance of trehalose, a known tracer for Asian dust, was found during the most severe dust storm event (4–5 May). Anhydrosugars, xylose, maltose, inositol and erythritol mostly existed in the fine mode, while other primary sugars and sugar alcohols dominated in coarse particles. Dust storm is a major source of organic compounds in the coarse particles, which induce a remarkable size shift to the coarse mode. Fine mode GMDs of biomass burning aerosols and primary biological aerosol particles were larger in the haze days than in the non-haze days, probably due to the stable meteorological conditions with higher humidity, which favor the condensation of organic matters onto aerosol particles, especially in the coarse mode.

Based on the tracer-based methods, the contributions of biomass burning, plant debris and fungal spores to OC were calculated with a percentage of 1.21–7.52%, 0.09–0.30% and 0.13–1.00%, respectively. The contribution of biomass burning-derived OC predominated in the fine mode throughout the year, although in the size range of 2.1–9.0 μm were not negligible. The contribution of plant debris-derived OC dominated in the coarse mode and decreased with the decreasing size. The contribution of fungal spores-derived OC dominated in the size range of 3.3–5.8 μm in summertime samples. Though the sources of plant debris and fungal spores prevailed in spring and summer, small contributions in the coarse mode were also





found during cold seasons due to long-range atmospheric transport from the southern regions. To better characterize sources and formation processes of haze particles and atmospheric chemistry in the North China Plain, further investigations on the size distributions of biogenic and anthropogenic SOA tracers are strongly needed in the future.

## 5   Supporting Information Available

Information on sampling parameters from April 2017 to January 2018 (Table S1). Linear correlation coefficients among anhydrosugars and OC, and with EC in Beijing aerosols from April 2017 to January 2018 (Table S2). Linear correlation coefficients among primary saccharides and sugar alcohols, and with levoglucosan in Beijing aerosols from April 2017 to January 2018 (Table S3). Geometric mean diameters (GMDs, mm) of anhydrosugars and other saccharides in the fine (<2.1 μm) and coarse (≥2.1 μm) mode particles in the Beijing (Table S4 and Table S5). Clusters of 3-day backward trajectories of air masses arriving at Beijing during dust storm events and haze periods from April 2017 to January 2018 (Fig. S1).

*Data availability*. The dataset for this paper is available upon request from the corresponding author (fupingqing@tju.edu.cn).

*Competing interests*. The authors declare that they have no conflict of interest.

*Author contributions.* PQF designed this research. Field campaigns were organized by PQF, SJH and YCL. Aerosol samples were collected by SJH. Laboratory analyses were performed by SFX, LJR, HR, and SJH. The manuscript was written by SFX and PQF with consultation and editing from all other authors.

*Acknowledgements.* This work was supported by the National Natural Science Foundation of China (Grant Nos. 41625014, 41581130024 and 41807303) and the State Key Joint Laboratory of Environment Simulation and Pollution Control (No. 18K02ESPCT).



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


**Table 1.** Abundances (ng m$^{-3}$) and seasonal variations in anhydrosugars and other saccharides in size-segregated aerosols in urban Beijing from April 2017 to January 2018. In total 16 sets of size-segregated samples were analysed; each set contains 9 samples.

| Compounds | Annual (N=16 sets) | | Spring (N=4) | | Summer (N=4) | | Autumn (N=4) | | Winter (N=4) | |
|---|---|---|---|---|---|---|---|---|---|---|
| | mean | range | mean | range | mean | range | mean | range | mean | range |
| *Anhydrosugars* | | | | | | | | | | |
| Galactosan | 27.6 | 5.09–108 | 19.8 | 5.97–29.7 | 8.71 | 5.09–15.2 | 35.6 | 15.3–54.4 | 46.3 | 8.68–108 |
| Mannosan | 46.8 | 8.41–168 | 33.3 | 12.4–50.7 | 14.2 | 8.41–27.7 | 66.6 | 29.5–107 | 72.9 | 16.4–168 |
| Levoglucosan | 365 | 80.1–1230 | 288 | 100–563 | 114 | 80.1–203 | 501 | 224–759 | 559 | 131–1230 |
| subtotal | 440 | 97.8–1500 | 341 | 118–642 | 137 | 97.8–246 | 603 | 269–920 | 678 | 250–1500 |
| L/M | 8.27 | 6.34–11.8 | 8.49 | 6.34–11.4 | 8.32 | 7.33–9.52 | 7.62 | 7.08–8.20 | 8.64 | 7.30–11.8 |
| *Sugar Alcohols* | | | | | | | | | | |
| Arabitol | 36.7 | 5.35–149 | 32.2 | 7.25–84.6 | 79.6 | 43.4–149 | 20.9 | 5.35–29.1 | 14.1 | 5.52–33.8 |
| Mannitol | 58.4 | 3.64–259 | 41.3 | 11.2–89.1 | 151 | 65.1–259 | 20.4 | 3.64–37.7 | 21.2 | 5.75–56.9 |
| Inositol | 21.8 | 5.05–69.2 | 41.5 | 20.7–69.2 | 14.1 | 7.50–28.5 | 19.2 | 12.0–26.9 | 12.4 | 5.05–26.0 |
| Erythritol | 21.8 | 3.16–44.1 | 9.92 | 3.16–15.8 | 30.2 | 13.8–44.1 | 22.3 | 11.1–33.0 | 24.9 | 10.7–41.6 |
| Subtotal | 139 | 29.8–418 | 125 | 56.9–143 | 275 | 155–418 | 82.8 | 32.1–118 | 72.6 | 29.8–158 |
| *Sugars* | | | | | | | | | | |
| Glucose | 162 | 38.9–502 | 255 | 107–502 | 234 | 184–279 | 71.2 | 38.9–93.3 | 86.4 | 60.0–139 |
| Sucrose | 267 | 30.8–1590 | 743 | 266–1590 | 114 | 30.8–275 | 73.7 | 32.6–115 | 137 | 87.7–209 |
| Fructose | 30.1 | 7.97–72.5 | 41.7 | 25.8–72.5 | 34.5 | 18.9–64.0 | 27.4 | 18.1–33.7 | 16.8 | 7.97–28.8 |
| Maltose | 5.65 | 2.06–16.9 | 5.14 | 3.24–8.62 | 4.21 | 2.15–8.63 | 6.77 | 3.69–9.96 | 6.50 | 2.06–16.9 |
| Xylose | 19.4 | 9.12–32.7 | 21.4 | 12.0–32.6 | 13.2 | 9.12–19.6 | 22.2 | 10.2–29.7 | 20.7 | 11.0–32.7 |
| Trehalose | 136 | 12.4–559 | 199 | 51.8–559 | 232 | 101–378 | 51.7 | 12.4–85.6 | 61.1 | 33.8–98.9 |
| Subtotal | 620 | 116–2330 | 1265 | 474–2330 | 632 | 356–1020 | 253 | 116–353 | 328 | 205–525 |

Note: The samples are divided into spring (April-May), summer (June-July), autumn (October-November) and winter (December-January).





**Table 2.** Abundances (ng m$^{-3}$) and seasonal variations in anhydrosugars and other saccharides in haze and non-haze days from April 2017 to January 2018.

| Compounds | Spring | | Summer | | Autumn | | Winter | |
|---|---|---|---|---|---|---|---|---|
| | dust-storm | non-haze | haze | non-haze | haze | non-haze | haze | non-haze |
| *Anhydrosugars* | | | | | | | | |
| Galactosan | 17.8 | 21.8 | 11.4 | 6.05 | 49.1 | 22.2 | 164 | 10.5 |
| Mannosan | 30.9 | 35.6 | 19.0 | 9.53 | 90.4 | 42.9 | 257 | 17.5 |
| Levoglucosan | 245 | 332 | 343 | 85.6 | 659 | 343 | 1880 | 175 |
| subtotal | 293 | 389 | 373 | 101 | 798 | 408 | 2310 | 203 |
| *Sugar Alcohols* | | | | | | | | |
| Arabitol | 56.3 | 8.10 | 111 | 47.9 | 28.0 | 13.9 | 45.0 | 5.67 |
| Mannitol | 68.5 | 14.0 | 74.0 | 228 | 33.3 | 7.48 | 36.7 | 6.40 |
| Inositol | 61.3 | 21.7 | 19.6 | 8.57 | 25.5 | 12.9 | 36.7 | 6.45 |
| Erythritol | 10.3 | 9.50 | 33.9 | 26.5 | 28.9 | 15.6 | 74.4 | 12.6 |
| Subtotal | 196 | 53.3 | 239 | 311 | 116 | 49.9 | 193 | 31.1 |
| *Sugars* | | | | | | | | |
| Glucose | 384 | 125 | 252 | 216 | 92.9 | 49.6 | 223 | 61.4 |
| Sucrose | 1160 | 328 | 184 | 44.7 | 99.4 | 47.9 | 343 | 101 |
| Fructose | 57.3 | 26.0 | 45.0 | 24.1 | 30.4 | 24.4 | 49.3 | 8.94 |
| Maltose | 4.35 | 5.93 | 6.07 | 2.35 | 8.37 | 5.16 | 21.7 | 2.16 |
| Xylose | 20.6 | 22.3 | 16.1 | 10.4 | 27.7 | 16.7 | 60.4 | 11.3 |
| Trehalose | 343 | 55.6 | 316 | 148 | 76.1 | 27.3 | 155 | 44.8 |
| Subtotal | 1970 | 563 | 819 | 446 | 335 | 171 | 852 | 230 |



**Table 3.** Abundance (ng m$^{-3}$) of anhydrosugars and other saccharides in the total, fine (<2.1 μm) and coarse (≥2.1 μm) particles in Beijing from April 2017 to January 2018.

| Compounds | Spring | | | Summer | | | Autumn | | | Winter | | |
|---|---|---|---|---|---|---|---|---|---|---|---|---|
| | Total | Coarse | Fine | Total | Coarse | Fine | Total | Coarse | Fine | Total | Coarse | Fine |
| *Anhydrosugars* | | | | | | | | | | | | |
| Galactosan | 19.8±11.6 | 8.41±3.55 | 11.4±9.50 | 8.71±4.43 | 4.73±3.32 | 4.03±1.19 | 58.9±34.6 | 7.64±3.83 | 51.3±30.9 | 46.3±46.6 | 5.86±5.05 | 40.5±41.5 |
| Mannosan | 33.3±19.7 | 9.72±3.55 | 23.5±18.6 | 14.3±9.05 | 4.72±4.20 | 9.62±4.85 | 101±49.3 | 8.97±4.26 | 92.3±45.4 | 72.9±71.8 | 7.40±5.43 | 65.5±66.3 |
| Levoglucosan | 288±205 | 71.5±25.3 | 217±201 | 114±60.0 | 40.6±41.0 | 73.8±19.9 | 751±340 | 73.0±28.9 | 679±312 | 559±501 | 64.4±46.0 | 494±457 |
| Subtotal | 341±233 | 89.7±31.7 | 252±227 | 137±73.4 | 50.0±48.5 | 87.4±25.7 | 912±424 | 89.6±36.7 | 822±388 | 678±619 | 77.6±56.1 | 600±565 |
| *Sugar Alcohols* | | | | | | | | | | | | |
| Arabitol | 32.2±36.2 | 30.9±36.5 | 1.34±0.81 | 79.6±47.8 | 78.0±46.8 | 1.70±1.03 | 28.0±4.74 | 24.5±3.92 | 3.56±0.85 | 14.1±13.4 | 11.7±11.8 | 2.34±1.69 |
| Mannitol | 41.3±35.8 | 39.0±36.2 | 2.25±1.04 | 151±92.6 | 147±91.8 | 3.48±2.29 | 33.7±19.0 | 30.5±18.5 | 3.22±1.09 | 21.2±24.1 | 19.0±23.5 | 2.22±1.06 |
| Inositol | 41.5±23.8 | 34.7±24.9 | 6.79±2.54 | 14.1±9.72 | 7.98±6.63 | 6.20±3.25 | 22.7±5.98 | 5.31±2.42 | 17.4±4.51 | 12.4±9.36 | 4.97±1.81 | 7.42±7.70 |
| Erythritol | 9.92±5.76 | 5.59±2.56 | 4.33±4.13 | 30.2±14.0 | 22.8±11.1 | 7.42±2.67 | 29.9±9.40 | 14.2±3.69 | 15.7±5.90 | 24.9±14.7 | 13.3±7.32 | 11.7±8.61 |
| Subtotal | 125±93.6 | 110±97.6 | 14.7±8.35 | 275±138 | 256±131 | 18.8±7.91 | 114±37.0 | 74.5±27.2 | 39.9±10.7 | 72.6±60.0 | 49.0±41.4 | 23.6±18.7 |
| *Sugars* | | | | | | | | | | | | |
| Glucose | 255±178 | 239±178 | 15.2±4.78 | 234±40.0 | 214±41.3 | 20.0±5.03 | 96.4±32.6 | 72.9±32.3 | 23.5±3.72 | 86.5±36.9 | 73.5±30.2 | 12.9±6.82 |
| Sucrose | 743±599 | 718±601 | 24.9±5.16 | 114±110 | 97.6±101 | 17.0±9.32 | 118±64.2 | 95.0±50.3 | 22.7±15.1 | 137±51.8 | 110±39.6 | 26.2±16.8 |
| Fructose | 41.7±21.9 | 35.6±21.4 | 6.03±2.48 | 34.6±20.1 | 25.3±19.0 | 9.32±2.87 | 30.1±2.79 | 16.3±3.34 | 13.8±2.96 | 16.8±9.71 | 11.3±6.20 | 5.47±3.95 |
| Maltose | 5.14±2.42 | 1.64±0.28 | 3.50±2.17 | 4.21±3.01 | 2.82±2.43 | 1.42±0.55 | 10.1±4.79 | 3.34±4.27 | 6.72±1.51 | 6.50±7.02 | 3.33±4.29 | 3.17±2.77 |
| Xylose | 21.4±9.79 | 8.73±2.83 | 12.7±9.70 | 13.3±4.49 | 7.27±2.71 | 6.01±1.97 | 27.8±4.21 | 5.03±1.39 | 22.8±4.74 | 20.7±11.1 | 5.53±1.40 | 15.2±10.5 |
| Trehalose | 199±242 | 189±228 | 10.8±14.2 | 232±116 | 228±114 | 3.90±2.47 | 73.3±24.6 | 69.1±22.6 | 4.29±2.17 | 61.1±27.3 | 56.4±25.7 | 4.64±2.13 |
| Subtotal | 1270±864 | 1190±863 | 73.1±23.3 | 632±283 | 575±269 | 57.7±18.6 | 355±125 | 262±106 | 93.8±21.6 | 328±141 | 260±103 | 67.6±37.5 |

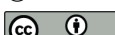



**Table 4.** Geometric mean diameters (GMDs, μm) of anhydrosugars and other saccharides in the total mode particles in Beijing from April 2017 to January 2018. GMD: logGMD = ($\sum$C$_i$log$D_{pi}$)/$\sum$C$_i$, where C$_i$ is the concentration of a compound in size stage i and $D_{pi}$ is the geometric mean diameter of particles collected on stage i.

| Compounds | Total | | | | | | | |
| --- | --- | --- | --- | --- | --- | --- | --- | --- |
| | Spring | | Summer | | Autumn | | Winter | |
| | Dust storm | Non-haze | Haze | Non-haze | Haze | Non-haze | Haze | Non-haze |
| *Anhydrosugars* | | | | | | | | |
| Galactosan | 1.47 | 1.04 | 1.38 | 1.06 | 0.94 | 0.80 | 0.72 | 0.66 |
| Mannosan | 1.15 | 0.83 | 1.07 | 0.82 | 0.84 | 0.76 | 0.68 | 0.65 |
| Levoglucosan | 1.12 | 0.62 | 1.15 | 0.80 | 0.82 | 0.58 | 0.64 | 0.57 |
| *Sugar Alcohols* | | | | | | | | |
| Arabitol | 9.94 | 5.54 | 4.90 | 4.09 | 3.48 | 2.58 | 3.09 | 3.09 |
| Mannitol | 9.63 | 4.86 | 4.90 | 4.08 | 3.46 | 2.59 | 3.58 | 3.46 |
| Inositol | 7.81 | 2.73 | 1.62 | 0.85 | 0.78 | 0.57 | 0.89 | 1.89 |
| Erythritol | 2.14 | 1.39 | 2.31 | 3.26 | 1.59 | 1.33 | 1.73 | 1.56 |
| *Sugars* | | | | | | | | |
| Glucose | 8.97 | 5.76 | 3.73 | 3.35 | 2.62 | 1.96 | 3.63 | 4.24 |
| Sucrose | 10.2 | 11.3 | 4.18 | 2.14 | 3.35 | 2.55 | 3.68 | 3.92 |
| fructose | 10.4 | 6.96 | 2.51 | 1.65 | 1.58 | 1.43 | 2.53 | 1.84 |
| Maltose | 0.65 | 0.70 | 1.97 | 0.89 | 0.70 | 0.65 | 1.54 | 0.82 |
| Xylose | 1.54 | 0.92 | 1.47 | 1.57 | 0.80 | 1.03 | 0.91 | 0.93 |
| Trehalose | 6.61 | 5.72 | 5.32 | 4.55 | 4.66 | 4.02 | 4.88 | 5.87 |





**Table 5.** Abundance of estimated primary OC (plant debris, fungal spores, biomass burning) and their contributions to OC in urban Beijing aerosols from April 2017 to January 2018.

| Compounds | Spring | | Summer | | Autumn | | Winter | |
|---|---|---|---|---|---|---|---|---|
| | mean | range | mean | range | mean | range | mean | range |
| *Abundance (unit: µg m⁻³)* | | | | | | | | |
| Plant debris OC | 0.34 | 0.19–0.66 | 0.31 | 0.24–0.37 | 0.09 | 0.05–0.12 | 0.11 | 0.08–0.18 |
| Fungal spore OC | 0.32 | 0.09–0.68 | 1.15 | 0.50–1.98 | 0.16 | 0.03–0.29 | 0.16 | 0.04–0.12 |
| Biomass burning OC | 3.51 | 1.22–6.87 | 1.39 | 0.98–2.48 | 6.11 | 2.75–9.26 | 6.81 | 2.67–14.9 |
| Sum of primary OC | 4.17 | 1.49–7.14 | 2.86 | 1.85–4.35 | 6.36 | 2.82–9.60 | 7.09 | 1.73–15.6 |
| *Contribution to OC (%)* | | | | | | | | |
| Plant debris OC | 0.30 | 0.14–0.63 | 0.30 | 0.17–0.45 | 0.09 | 0.08–0.10 | 0.16 | 0.12–0.20 |
| Fungal spore OC | 0.23 | 0.10–0.35 | 1.00 | 0.63–1.81 | 0.13 | 0.04–0.23 | 0.17 | 0.09–0.32 |
| Biomass burning OC | 3.67 | 0.84–8.37 | 1.21 | 0.89–1.40 | 5.51 | 4.22–6.35 | 7.52 | 3.40–11.2 |
| Sum of primary OC | 4.20 | 1.26–8.71 | 2.51 | 2.01–2.97 | 5.73 | 4.34–6.54 | 7.85 | 3.67–11.6 |



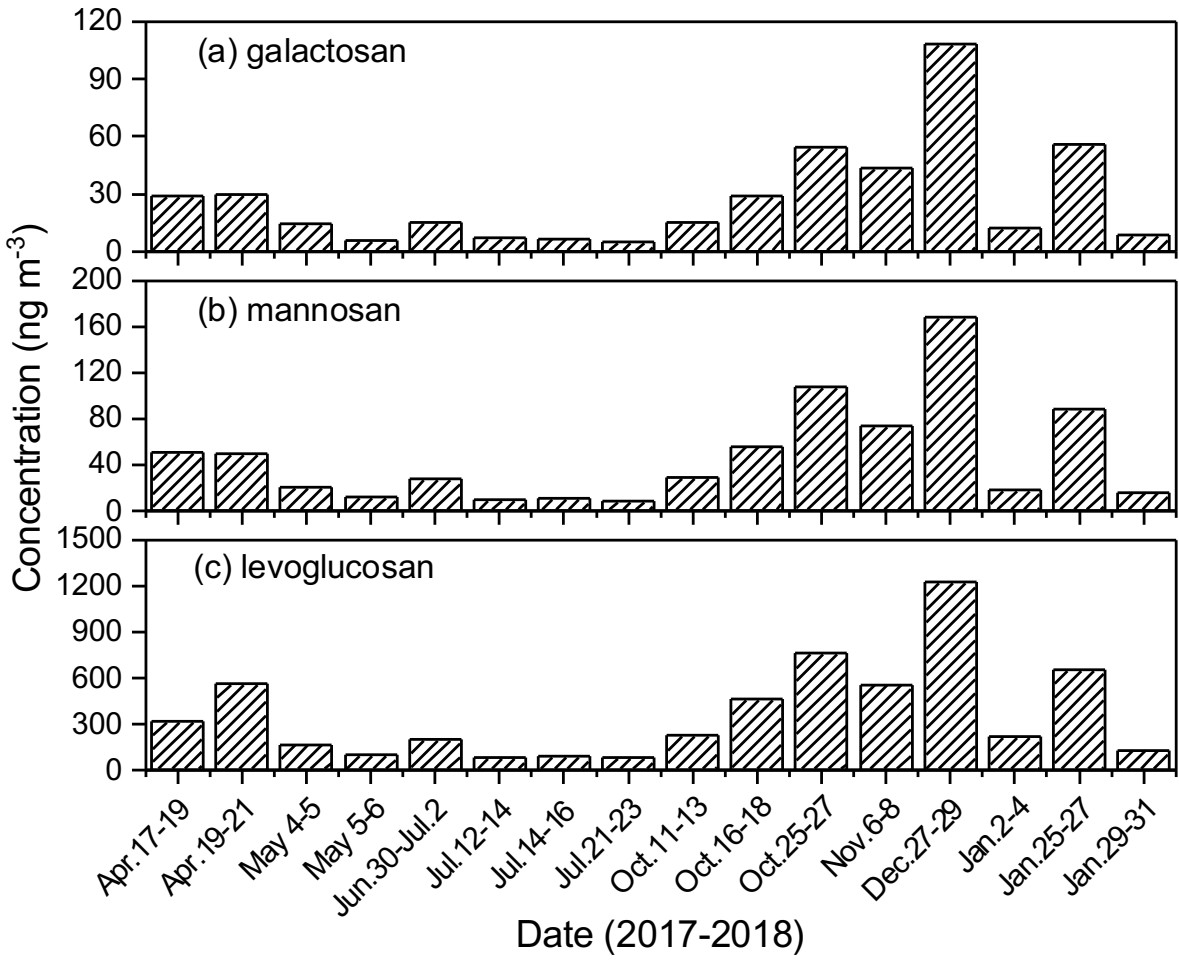

**Figure 1.** Seasonal trends (a, b, c) of anhydrosugars in urban Beijing aerosols from April 2017 to January 2018.





**Figure 2.** Temporal variations in the concentration ratios of L/M (levoglucosan to mannosan), M/G (mannosan to galactosan), L/OC (levoglucosan to organic carbon) and L/EC (levoglucosan to elemental carbon) for urban aerosols in Beijing.



**Figure 3.** The concentration ratios of L/M, M/G, L/OC and L/EC according to particle size for urban aerosols in Beijing during April 2017 to January 2018.






**Figure 4.** Temporal variation in primary saccharides and sugar alcohols in urban Beijing aerosols from April 2017 to January 2018.







**Figure 5.** Size distributions of levoglucosan and its isomers, mannosan and galactosan, in urban aerosols in Beijing.







**Figure 6.** Size distributions of glucose, fructose, sucrose, mannitol and arabitol in urban aerosols in Beijing.





**Figure 7.** Size distributions of trehalose, xylose, maltose, inositol and erythritol in urban aerosols in Beijing.



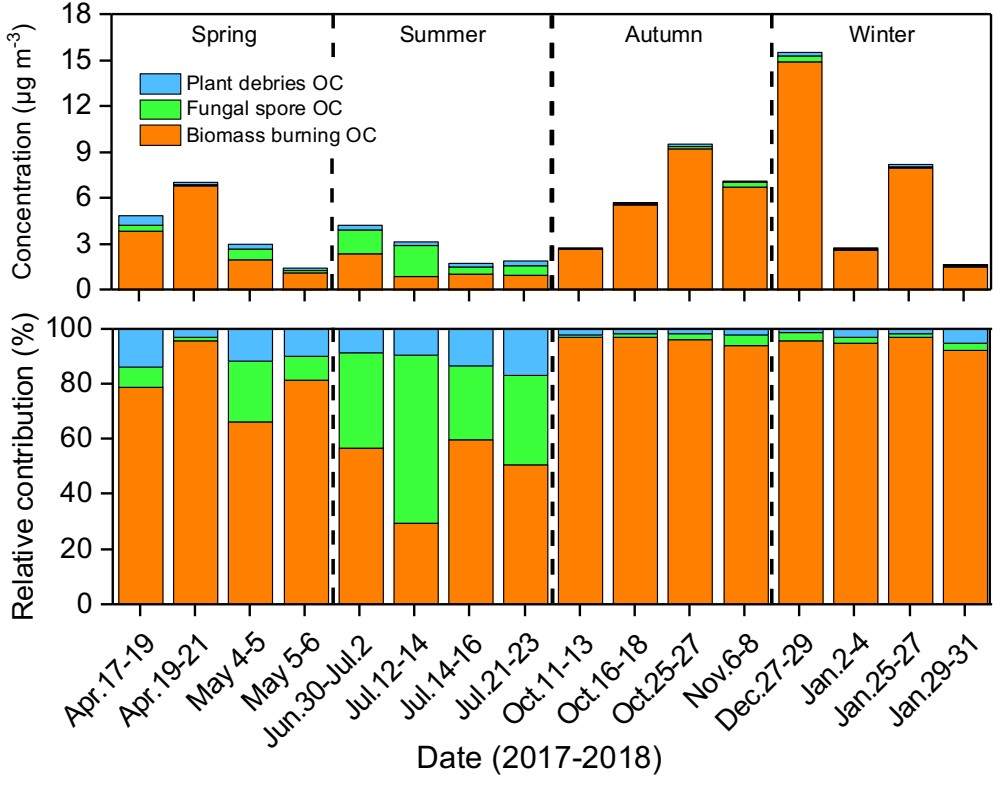

**Figure 8.** Monthly variation in plant debris-derived OC, fungal spores-derived OC, and BB-derived OC in urban aerosols in Beijing and relative contributions of these primary OC.





**Figure 9.** Concentrations of OC derived from plant debris, fungal spores and biomass burning in different size ranges of urban aerosols in Beijing.





**Figure 10.** Relative contributions of plant debris-derived OC, fungal spores-derived OC and BB-derived OC in different size ranges of urban aerosols in Beijing.