# Peer review of "Molecular markers of biomass burning and primary biological aerosols in urban Beijing: Size distribution and seasonal variation"

_Atmospheric Chemistry and Physics, 2019_

## Referee Comment (RC1) · Anonymous Referee #1 · 6 Nov 2019

The manuscript by Xu et al. investigated molecular markers of primary biomass burning and biological aerosols during different seasons in Beijing, with focuses on size distribution and seasonal variation. Four sets of ambient aerosol samples were collected for each season using a nine-stage cascade impactor sampler, leading to a total of sixteen sets of samples for the entire measurement period. The samples were analyzed for anhydrosugars, sugar alcohols and sugars. Based on the measurement results, the authors discussed the abundances, seasonal variations and size distributions of these compounds, then estimated the contributions of biomass burning, plant debris and fungal spores to OC. In principle, the topic of this manuscript falls within the scope of Atmospheric Chemistry and Physics. However, I could not support its

publication due to the following concerns.

1. There have been many previous studies investigating the concentrations of saccharides in Beijing aerosol (e.g., Liang et al., Chemosphere, 2016, 150, 365-377). Although these studies typically relied on PM2.5 and/or PM10 rather than size resolved samples, they generally covered much more sampling days than the present study, and therefore much more representative when discussing the abundances and seasonal variations of saccharides as well as when estimating the contributions of biomass burning, plant debris and fungal spores to OC. Compared to the results from previous studies, are there any new findings in Sections 3.1, 3.2 and 3.4? Maybe the authors should focus on the size distributions of saccharides, which may be able to differ the present study from previous ones.

2. It is completely unclear how the haze, non-haze and dust-storm periods (which were frequently mentioned throughout the manuscript) were identified.

3. Considering the formation and evolution processes of haze events in Beijing (which could be fast; e.g., Sun et al., Sci. Rep., 2016, 6, 27151), it is questionable whether the so-called haze samples were representative (recalling that only four sets of samples were collected for each season).

4. Please clarify why dust storm is a major source of OC in coarse particles. This point was presented as a conclusion but was not clearly explained in the manuscript.

5. A minor point. Page 6, Line 28. Please check the two ratios cited here.

––––––––––––––––––––––––––––––

---

## Referee Comment (RC2) · Anonymous Referee #2 · 20 Nov 2019

This paper identified and quantified the Molecular markers of biomass burning and primary biological aerosols. Atmospheric concentrations, seasonal variations and size distributions of anhydrosugars and sugar alcohols were investigated, the analysis and interpretation of the results are overall fair. The paper presents useful information about the organic aerosols. However, some additional information is still necessary for the readers to better understand this work. Comments: 1. Three anhydrosugars, six primary saccharides and four sugar alcohols should be introduced clear in the section 1, such as the sources, property. 2. In the section 2.1, a brief introduction to the surrounding environment of sampling sites should be provided, and the details about the sampling time should be described, including start time, end time, and duration. 3.

There are only four cases (less than 10 days) in each season, more field measurement estimation could decrease the influence produced by accidental elements, making the results more convincing. 4. Lots of ratios of individual anhydrosugars were showed in section 3.1.2, the result were almost consistent with previous results, the conclusion should be summarized and classified, so that the author can better understand. 5. The size distribution of Anhydrosugars, Primary saccharides and Sugar alcohols was analyzed in Secction 3.3, a size shift towards large particles and large GMDs in the fine fraction (<2.1 $\mu$m) was detected during the hazy days. The author ascribes them to higher humidity, but did not analyze the impact of RH on size distribution. Please give more discussion.

---

## Author Comment (AC1) · 17 Jan 2020

**Responses to Reviewer #1**

We appreciate the thoughtful comments and suggestions from the reviewers, which greatly improved the quality of our manuscript. The point-by-point responses to these comments are listed below with referee's comments in black and our responses in blue.

Reviewer #1 (Formal Review for Author (shown to author)):

The manuscript by Xu et al. investigated molecular markers of primary biomass burning and biological aerosols during different seasons in Beijing, with focuses on size distribution and seasonal variation. Four sets of ambient aerosol samples were collected for each season using a nine-stage cascade impactor sampler, leading to a total of sixteen sets of samples for the entire measurement period. The samples were analyzed for anhydrosugars, sugar alcohols and sugars. Based on the measurement results, the authors discussed the abundances, seasonal variations and size distributions of these compounds, then estimated the contributions of biomass burning, plant debris and fungal spores to OC. In principle, the topic of this manuscript falls within the scope of Atmospheric Chemistry and Physics. However, I could not support its publication due to the following concerns.

Response:

We appreciated the valuable comments from the reviewer. Organic molecular characterization of urban aerosols has been conducted comprehensively during the past decade. However, very limited studies have been conducted for size-segregated aerosol samples. Thus, we believe that our detailed characterization of size distributions of saccharides in urban Beijing provides useful information on the biomass burning and fungal spore tracers and their patterns of size distributions and GMDs for the first time, which is worth publication in ACP.

1、There have been many previous studies investigating the concentrations of saccharides in Beijing aerosol (e.g., Liang et al., Chemosphere, 2016, 150, 365-377). Although these studies typically relied on $PM_{2.5}$ and/or $PM_{10}$ rather than size resolved samples, they generally covered much more sampling days than the present study, and therefore much more representative when discussing the abundances and seasonal variations of saccharides as well as when estimating the contributions of biomass burning, plant debris and fungal spores to OC.

Response:

We appreciated the valuable comments from the reviewer. We know that there are excellent studies focusing on the concentrations of saccharides in Beijing aerosol, while there is still a

lack of knowledge on the size distribution of these organic species. In fact, most of the previous studies use high performance anion-exchange chromatography (HPAEC) (e.g. Liang et al., 2016). Here, we measured anhydrosugars, primary saccharides and sugar alcohols using GC/MS in this study. Generally, many studies were carried out based on dozens of samples by GC/MS (Fu et al., 2008; Li et al., 2018; Wan et al., 2019; Wang et al., 2006). Though we could not take into full consideration the sampling days, each set of our samples analyzed in our study are representative and adequate when discussing the abundances and seasonal variations of saccharides as well as when estimating the contributions of biomass burning, plant debris and fungal spores to OC. Each sample set corresponded to specific meteorological condition, which are listed in Table S1.

Compared to the results from previous studies, are there any new findings in Sections 3.1, 3.2 and 3.4? Maybe the authors should focus on the size distributions of saccharides, which may be able to differ the present study from previous ones.

Response:

In Section 3.1, we give a detailed description about the L/M, M/G, L/OC, and L/EC ratios according to particle size. New findings were as followed:

(1) Higher L/M and M/G values were observed in the fine mode, which ascribed to the difference of burning substrates. Hardwood was potentially the burning substrates in the fine mode while mixture impact of hardwood and softwood burning accounted for the relatively lower L/M ratios in the coarse mode. Higher M/G ratios in the fine mode implied the increasing contributions from crop straws burning, especially in haze days.

(2) Dust storms could induce a high L /EC ratio in the coarse mode because of coarse particles brought by dust storms and/or road dust resuspension. While decreased concentrations of EC in the coarse mode in dust storms implying that EC may derived of local emissions rather than long distance transportation of dust.

In Section 3.4, we calculated the contributions of OC from BB, plant debris and fungal spores in terms of particle size. The contributions of BB-OC were different in the fine and coarse mode. BB-OC dominated in the fine mode (>90%) and the contribution of BB-OC in the size range of 2.1–9.0 μm were with a proportion over 60%. While in a larger size range, the relative contribution were below 60% in most periods. Plant debris-derived OC and fungal spores-derived OC mostly existed in the coarse mode. However, both of them were present in the fine mode, especially in spring and summer. Though the contribution of plant debris and

fungal spores in the whole range were insignificant in cold seasons, their relative contribution in the coarse mode were comparatively high.

2、It is completely unclear how the haze, non-haze and dust-storm periods (which were frequently mentioned throughout the manuscript) were identified.

Response:

Thanks for the reviewer for pointing out the missing data. Such information on the meteorological parameters during each sampling period is added in Table S1. The two sets (11–19 April and 4–5 May) were collected in dust storm days, the following sets (30 June–2 July, 12–14 July, 25–27 October, 6–8 November, 27–29 December and 25–27 January) were affected by haze. And the rest sets were for non-haze days.

3、Considering the formation and evolution processes of haze events in Beijing (which could be fast; e.g., Sun et al., Sci. Rep., 2016, 6, 27151), it is questionable whether the so-called haze samples were representative (recalling that only four sets of samples were collected for each season).

Response:

In many previous studies, total suspended particles (TSP) samples were collected by a high-volume sampler, with an operating flow rate of 1.00 $m^3$/min, approximately (Chen et al., 2013; Li et al., 2018; Wan et al., 2019). While in this study, all samples were collected using a nine-stage cascade impactor sampler (Andersen, U.S.A.) at a flow rate of 25.8 L/min from April 2017 to January 2018. Compared to the high-volume sampler, the flow rate of nine-stage cascade impactor sampler is much lower. If the sampling durations were too short, the circumstance of the concentrations of size-resolved samples below the detection line will occur. As a result, we had to prolong the sampling time to guarantee the validity of samples, especially for the non-haze days. Former studies found that the formation and evolution processes of haze events could be fast, sometimes happened even less in one day (Sun et al., 2014; Yang et al., 2015). To completely encompass the durations of the rapid formation of haze events and the evolution of secondary organic aerosols (which not discussed in this manuscript), we considered 2 to 3 days as a reasonable sampling period. In addition, such sampling period is necessary to collect enough particles for organic analysis.

4、Please clarify why dust storm is a major source of OC in coarse particles. This point was presented as a conclusion but was not clearly explained in the manuscript.

Response:

Thanks for the suggestion. In this manuscript, we presented "Dust storm is a major source of organic compounds in the coarse particles, which induce a remarkable size shift to the coarse mode". This description may be a little inappropriate. We corrected the conclusion in the revised manuscript as "Dust storm greatly enhance organic aerosol concentrations and induce a remarkable size shift towards coarse sizes (see Page 15, Lines 9–10). There are several reasons for this conclusion. Firstly, in general, the concentrations of most primary saccharides sugar alcohols in each impactor stage during dust storms (17–19 April and 4–5 May) were higher than those of non-haze days (19–21 April and 5–6 May), especially for the coarse particle fraction (Figure 6–7). As for anhydrosugars and related sugars, their concentrations in the coarse mode in dust storm were higher, too. Such phenomenon could be probably attributed to strongly windblown mass coarse dust derived from large scale resuspension of dust from crustal, soil, roads or other unpaved areas, along with long-range transport of particles from north and northwest desert regions. Previous studies found that elevated concentrations of trehalose, mannitol and arabitol are generally related to resuspended soil and the outflow of dust storms (Liang et al., 2013; Rogge et al., 2007). Secondly, the GMDs of the total size range and the coarse mode particles in dust storm were larger than non-haze and haze days (Table 4–S5). Some species, such as arabitol, mannitol and inositol, their GMDs associated with coarse particles in dust storm presented a significantly increase, again suggesting the effect of dust storms on the aerosol particle size. Wang et al. (2013) also found that dust storms could act as a major source of coarse particulate matter.

5、A minor point. Page 6, Line 28. Please check the two ratios cited here.

Response:

Thanks. We have corrected the mistake in the revised manuscript (see Page 6, Line 25–26). The revised content is as followed:

"The M/G ratios during all the periods were in a range of 1.35–2.08 with an average 1.70 (Fig. 2b). The M/G ratios maximized in autumn (1.68–1.97, 1.88) and minimized in summer (1.35–1.82, 1.59)."

**Table S1.** Information on the weather conditions during each of the sampling periods from April 2017 to January 2018.

| Year | Season | Sampling period | Duration (min) | T (°C)[a] | RH (%)[b] | WS[c] | PM$_{2.5}$ | PM$_{10}$ | Weather conditions |
|------|--------|-----------------|----------------|-----------|-----------|-------|------------|-----------|--------------------|
| 2017 | spring | 17–19 Apr. | 2880 | 18.2 | 32.3 | 5 | 174 | 124 | dust storm |
| | | 19–21 Apr. | 2887 | 14.7 | 46.2 | 3 | 78.3 | 84.6 | non-haze |
| | | 4–5 May | 1364 | 20.1 | 33.4 | 6 | 501 | 656 | dust storm |
| | | 5–6 May | 1954 | 17.6 | 25.2 | 5 | 131 | 125 | non-haze |
| | summer | 30 Jun.–2 Jul. | 2862 | 29.3 | 70.5 | 2 | 143 | 103 | haze |
| | | 12–14 Jul. | 2854 | 31.5 | 76.7 | 1 | 89.2 | 75.9 | haze |
| | | 14–16 Jul. | 2900 | 29.0 | 58.3 | 2 | 65.5 | 62.1 | non-haze |
| | | 21–23 Jul. | 2843 | 25.2 | 64.2 | 3 | 42.7 | 36.3 | non-haze |
| | Autumn | 11–13 Oct. | 2877 | 12.3 | 64.2 | 2 | 38.1 | 32.4 | non-haze |
| | | 16–18 Oct. | 2900 | 13.1 | 69.1 | 2 | 78.0 | 68.7 | non-haze |
| | | 25–27 Oct. | 2865 | 11.8 | 81.3 | 1 | 183 | 120 | haze |
| | | 6–8 Nov. | 2887 | 9.36 | 72.4 | 2 | 146 | 91.7 | haze |
| 2017-2018 | Winter | 27–29 Dec. | 2781 | -2.34 | 75.2 | 1 | 137 | 140 | haze |
| | | 2–4 Jan. | 2757 | -2.68 | 32.4 | 3 | 33.7 | 32.1 | non-haze |
| | | 25–27 Jan. | 2858 | -8.13 | 41.6 | 1 | 82.2 | 59.6 | haze |
| | | 29–31 Jan. | 2835 | -1.95 | 22.4 | 2 | 47.5 | 49.5 | non-haze |

[a]temperature (T); [b]relative humidity (RH); [c]wind scale (WS).

References:

Chen, J., Kawamura, K., Liu, C. Q., and Fu, P.: Long-term observations of saccharides in remote marine aerosols from the western North Pacific: A comparison between 1990–1993 and 2006–2009 periods, Atmos. Environ., 67, 448-458, doi:10.1016/j.atmosenv.2012.11.014, 2013.

Fu, P., Kawamura, K., Okuzawa, K., Aggarwal, S. G., Wang, G., Kanaya, Y., and Wang, Z.: Organic molecular compositions and temporal variations of summertime mountain aerosols over Mt. Tai, North China Plain, J. Geophys. Res., 113, doi:10.1029/2008jd009900, 2008.

Li, L., Ren, L., Ren, H., Yue, S., Xie, Q., Zhao, W., et al.: Molecular characterization and seasonal variation in primary and secondary organic aerosols in Beijing, China, J. Geophys. Res., Atmospheres, 123, 12,394–12,412, doi:10.1029/2018JD028527, 2018.

Liang, L. L., Engling, G., Du, Z. Y., Cheng, Y., Duan, F. K., Liu, X. Y., and He, K. B.: Seasonal variations and source estimation of saccharides in atmospheric particulate matter in Beijing, China. Chemosphere. 150, 365–77, 2016.

Sun, Y.,Qi Jiang   Wang Z., Fu P., Li J., Yang T., Yin Y.: Investigation of the sources and evolution processes of severe haze pollution in Beijing in January 2013, J. Geophys. Res., 119, 4380–4398, doi:10.1002/2014JD021641, 2014.

Wan, X., Kang, S., Rupakheti, M., Zhang, Q., Tripathee, L., Guo, J., Chen, P., Rupakheti, D., Panday, A. K., Lawrence, M. G., Kawamura, K., and Cong, Z.: Molecular characterization of organic aerosols in the Kathmandu Valley, Nepal: insights into primary and secondary sources, Atmos. Chem. Phys., 19, 2725–2747, doi:10.5194/acp-19-2725-2019, 2019.

Wang, G., Kawamura, K., Lee, S., Ho, K., Cao, J. J.: Molecular, seasonal, and spatial distributions of organic aerosols from fourteen Chinese cities, Environ. Sci. Technol., 40, 4619–4625, doi:10.1021/es060291x, 2006.

Yang, Y., Liu, X., Qu, Y., An, J., Jiang, R., Zhang, Y., Ma, Q.: Characteristics and formation mechanism of continuous hazes in China: a case study during the autumn of 2014 in the North China Plain, Atmos. Chem. Phys. 15, 8165–8178, doi:10.5194/acp-15-8165-2015, 2015.

---

## Author Comment (AC2) · 17 Jan 2020

**Responses to Reviewer #2**

We are grateful to the reviewer for the thoughtful comments and suggestions, which greatly improved the quality of our manuscript. Below we make a point-by-point response to these comments. According to editor's requirement, the responses are structured in the following sequence: (1) comments from the referee is in black, (2) our responses are in blue.

Reviewer #2 (Formal Review for Author (shown to author)):

This paper identified and quantified the Molecular markers of biomass burning and primary biological aerosols. Atmospheric concentrations, seasonal variations and size distributions of anhydrosugars and sugar alcohols were investigated, the analysis and interpretation of the results are overall fair. The paper presents useful information about the organic aerosols. However, some additional information is still necessary for the readers to better understand this work.

Comments:

1. Three anhydrosugars, six primary saccharides and four sugar alcohols should be introduced clear in the section 1, such as the sources, property.

Response:

Thanks for the reviewer's suggestions. Before analysis, we have gave an introduction about all anhydrosugars, primary saccharides and sugar alcohols in Sections 3.1 and 3.2. Hence, we added the following sentences in the revised manuscript as a supplemental description in Section 1.

"Amongst sugar alcohols, mannitol and arabitol are the primary ones" (see Page 2, Line 20).

"Sparse data about erythritol and inositol can be found in the literature. Recently, these two species were reported as having similar sources as other sugar alcohols, such as arabitol and mannitol" (see Page 2, Line 23).

"On the basis of difference of burning materials (e.g. deciduous leafs, grass), BB is classified as open-field burning in forests, savannas, croplands and residential heating and cooking (Akagi et al., 2011; Yan et al., 2006)" (see Page 2, Line 25).

"Because levoglucosan is chemically stable in the air with no decay over 10 days" (see Page 3, Line 1).

"The levoglucosan to mannosan (L/M) ratio varies for aerosols generated by burning of hardwood, softwood and agricultural waste, with a ratio of 13–27, 2.5–5.8 and 25–55.7, respectively" (see Page 3, Line 2–3).

2. In the section 2.1, a brief introduction to the surrounding environment of sampling sites should be provided, and the details about the sampling time should be described, including start time, end time, and duration.

Response:

We have added the following paragraph in the revised manuscript (see Page 3, Line 18). "Institute of Atmospheric Physics (IAP), Chinese Academy of Sciences locate on the north side of Beijing city, which is considered be a representative urban urea for a mixed district of teaching, residential and commercial areas. Urban aerosol samples collected here are influenced by anthropogenic (transport emissions, cooking operations etc.) and natural sources (soil dust, plant debris, microorganisms etc.)."

The details about the sampling time (start time, end time, and duration) were presented in Table S1.

3. There are only four cases (less than 10 days) in each season, more field measurement estimation could decrease the influence produced by accidental elements, making the results more convincing.

Response:

Thanks for the reviewer for pointing this deficiency. Offline experiments allow investigators to repeat the process under controlled conditions to analyze the emission process, influencing factors and mechanisms (Chen et al., 2012; Zhang et al., 2011). The concentrations of organic tracers, such as compounds analyzed in our study were measured offline using GC/MS and OC/EC were also determined offline using a thermal/optical carbon analyzer (model RT-4, Sunset Laboratory Inc., USA). While field measurement are preferable and realistic since the field studies can reflect the real process at random conditions (Roden et al., 2006). It is realized that more online studies are required in our study. During sampling, we documented basic information about meteorological data, such as relative humidity, wind scale and so on simultaneously (see Table S1). These field measurement do support our conclusion. For example,

information about PM2.5 and PM10 defined the weather conditions precisely; higher relative humidity found in haze days do prefer hygroscopic growth, coagulation and/or condensation fine particle, resulting lager GMDs in the fine mode.

4. Lots of ratios of individual anhydrosugars were showed in section 3.1.2, the result were almost consistent with previous results, and the conclusion should be summarized and classified, so that the author can better understand.

Response:

Thanks for the reviewer's suggestion. We adjusted the order of the paragraphs and some sentences in Section 3.1.2 to arrange a clear and logical order. The first three paragraphs described the features of ratios of individual anhydrosugars and OC/EC, respectively. A new paragraph was added (the fourth paragraph) to give some comparison with other studies. The rest paragraphs were about ratios according to according to particle size.

The fourth paragraph is as followed:

"The L/M, L/OC and L/EC ratios calculated in this study were compared with those reported for other Asian cities in recent studies. The L/M ratios showed different seasonal and spatial distributions as city changed. Those reported in Nepal (16.3±5.96) were much higher than our study (Wan et al., 2019). Study in Okinawa found a L/M ratio of 10.7±6.1 and exhibited a similar seasonal trend to our study (Zhu et al., 2015a). Lower ratio of L/M suggested that more burning substrates were softwood in Beijing. While in previous studies in Beijing, lower ratios L/M in winter were observed (Cheng et al., 2013). Higher ratio of L/M in our study may ascribe to the reduced use of coal in recent years in Northern China. Coal combustion is also a source of levoglucosan, leading to a lower ratio of L/M (Yan et al., 2018). In comparison to annual mean L/OC ratios in Guangzhou and Zhaoqing in the PRD region ($10.8×10-3$ and $27.5×10-3$, respectively) (Ho et al., 2014), the lower values of L/OC in Beijing were potentially associated with the less extensive burning activities under the prohibition of open agricultural residue burning. Though indicating a similar seasonal trend to Okinawa, the L/EC ratios in Beijing were much higher than those in Okinawa. Besides the more release of levoglucosan in Beijing, such result may also imply that EC was associated with other sources, such local transport emission and coal combustion, while EC in Okinawa might represent a regional background level (Zhu et al., 2015a)."

5. The size distribution of Anhydrosugars, Primary saccharides and Sugar alcohols was analyzed in Section 3.3, a size shift towards large particles and large GMDs in the fine fraction (<2.1 m) was detected during the hazy days. The author ascribes them to higher humidity, but did not analyze the impact of RH on size distribution. Please give more discussion.

Response:

Thanks. Herner et al (2006) found that the particle size can be modified by chemical reactions, condensation/evaporation, coagulation with other particles, and activation during high humidity, wet and dry deposition. Anhydrosugars, primary saccharides and sugar alcohols are important parts of atmospheric primary organic aerosols (POA), which are highly water-soluble compared with fossil fuel derived particles (Reid et al., 2005). Larger GMDs of primary carbon particles in the fine mode in haze days than in non-haze days have been observed in previous and our studies. One reason for that is enhanced hygroscopic growth of the airborne particles under relatively higher humidity. As exhibited in Table S1, haze days were characterized by higher relative humidity in our study. Wang et al (2011) found that all fine mode of WSOC showed a larger GMD in the hazy days. Kang et al (2016) also discovered that the fine mode GMDs of aliphatic hydrocarbons generally larger in haze samples due to higher relative humidity. In addition, the increase concentrations of airborne aerosols could increase the overall coagulation rate, resulting in faster rate of collisions between particles and formation of larger particles (Herner et al., 2006). Table 2 showed that the concentrations of anhydrosugars, primary saccharides and sugar alcohols were much higher in haze days, which responsible for the larger GMDs in the fine mode and a size shift towards large particles.

**References:**

Chen Y., Roden C., Bond T.: Characterizing biofuel combustion with patterns of real-time emission data (PaRTED), Environ. Sci. Technol., 46, 6110–6117, doi: 10.1021/es3003348, 2012.

Cheng, Y., Engling, G., He, K. B., Duan, F. K., Ma, Y. L., Du, Z. Y., Liu, J. M., Zheng, M., and Weber, R. J.: Biomass burning contribution to Beijing aerosol, Atmos. Chem. Phys., 13, 7765-7781, doi:10.5194/acp-13-7765-2013, 2013.

Herner, J. D., Ying, Q., Aw, J., Gao, O., Chang, D. P., Kleeman, M. J.: Dominant mechanisms that shape the airborne particle size and composition distribution in central California, Aerosol Sci. Technol., 40, 827-844, doi:10.1080/02786820600728668, 2006.

Kang M., Fu P., Aggarwal S., Size distributions of n-alkanes, fatty acids and fatty alcohols in springtime aerosols from New Delhi, India, Environ. Pollut., 219, 957-966, doi:10.1016/j.envpol.2016.09.077, 2016.

Reid, J.S., Koppmann, R., Eck, T.F., Eleuterio, D.P.:A review of biomass burning emissions part II: intensive physical properties of biomass burning particles. Atmos. Chem. Phy. 5, 799–825, doi:10.5194/acp-5-799-2005, 2005.

Roden C., Bond T., Conway S., Pinel A.: Emission factors and real-time optical properties of particles emitted from traditional wood burning cookstoves, Environ. Sci. Technol., 40, 6750–6757,doi:10.1021/es052080i, 2006.

Wan, X., Kang, S., Rupakheti, M., Zhang, Q., Tripathee, L., Guo, J., Chen, P., Rupakheti, D., Panday, A. K., Lawrence, M. G., Kawamura, K., and Cong, Z.: Molecular characterization of organic aerosols in the Kathmandu Valley, Nepal: insights into primary and secondary sources, Atmos. Chem. Phys., 19, 2725–2747, doi:10.5194/acp-19-2725-2019, 2019.

Wang, G., Chen, C., Li, J., Zhou, B., Xie, M., Hu, S., Kawamura, K., and Chen, Y.: Molecular composition and size distribution of sugars, sugar-alcohols and carboxylic acids in airborne particles during a severe urban haze event caused by wheat straw burning, Atmos. Environ., 25 45, 2473-2479, doi:10.1016/j.atmosenv.2011.02.045, 2011.

Yan, C., Zheng, M., Sullivan, A.P., Shen, G., Chen, Y., Wang, S., Zhao, B., Cai, S., Desyaterik, Y., Li, X., Zhou, T., Gustafsson, O., Collett, J.L., Jr., 2018, Residential Coal Combustion as a Source of Levoglucosan in China, Environ. Sci. Technol., 52, 1665-1674, doi:10.1021/acs.est.7b05858, 2017.

Zhang H., Hu D., Chen J., Ye X., Wang S., Hao J., Wang L., Zheng R., An Z.: Particle size distribution and polycyclic aromatic hydrocarbons emissions from agricultural crop residue burning, Environ. Sci. Technol., 45, 5477–5482, doi:10.1021/es1037904, 2011.

Zhu, C., Kawamura, K., Kunwar, B.: Effect of biomass burning over the western North Pacific Rim: wintertime maxima of anhydrosugars in ambient aerosols from Okinawa, Atmos. Chem. Phys., 15, 1959–1973, doi:10.5194/acp-15-1959-2015, 2015a.

---

## Author Comment (AC3) · 17 Jan 2020

A list of all relevant changes made in the manuscript:

1. Information on the meteorological parameters during each of the sampling periods in detail are added and presented in Table S1 in supplementary information.

2. Extra description are provided about anhydrosugars, primary saccharides and sugar alcohols in Section 1 in the revised manuscript.

3. Some missing introductions and information are added in the section 2.1.

4. Ratios of individual anhydrosugars and OC/EC are summarized and classified in

[Figure]

Section 3.1. Errors about L/M ratio are corrected in the revised manuscript.

5. Explanation about "Dust storm greatly enhance organic aerosol concentrations and induce a remarkable size shift towards coarse sizes" are added in the revision.

---

## Author Response (AR1)

**Authors' Responses**

We are grateful to the thoughtful comments and suggestions from both reviewers, which greatly improved the quality of our manuscript. The point-by-point responses to these comments are listed below with referees' comments in black and our responses in blue.

**Responses to Reviewer #1**

Reviewer #1 (Formal Review for Author (shown to author)):

The manuscript by Xu et al. investigated molecular markers of primary biomass burning and biological aerosols during different seasons in Beijing, with focuses on size distribution and seasonal variation. Four sets of ambient aerosol samples were collected for each season using a nine-stage cascade impactor sampler, leading to a total of sixteen sets of samples for the entire measurement period. The samples were analyzed for anhydrosugars, sugar alcohols and sugars. Based on the measurement results, the authors discussed the abundances, seasonal variations and size distributions of these compounds, then estimated the contributions of biomass burning, plant debris and fungal spores to OC. In principle, the topic of this manuscript falls within the scope of Atmospheric Chemistry and Physics. However, I could not support its publication due to the following concerns.

Response:

We appreciated the valuable comments from the reviewer. Organic molecular characterization of urban aerosols has been conducted comprehensively during the past decade. However, very limited studies have been conducted for size-segregated aerosol samples. Thus, we believe that our detailed characterization of size distributions of saccharides in urban Beijing provides useful information on the biomass burning and fungal spore tracers and their patterns of size distributions and GMDs for the first time, which is worth publication in ACP.

1、There have been many previous studies investigating the concentrations of saccharides in Beijing aerosol (e.g., Liang et al., Chemosphere, 2016, 150, 365-377). Although these studies typically relied on $PM_{2.5}$ and/or $PM_{10}$ rather than size resolved samples, they generally covered much more sampling days than the present study, and therefore much more representative when discussing the abundances and seasonal variations of saccharides as well as when estimating the contributions of biomass burning, plant debris and fungal spores to OC.

Response:

We appreciated the valuable comments from the reviewer. We know that there are excellent studies focusing on the concentrations of saccharides in Beijing aerosol, while there is still a lack of knowledge on the size distribution of these organic species. In fact, most of the previous studies use high performance anion-exchange chromatography (HPAEC) (e.g. Liang et al., 2016). Here, we measured anhydrosugars, primary saccharides and sugar alcohols using GC/MS in this study. Generally, many studies were carried out based on dozens of samples by GC/MS (Fu et al., 2008; Li et al., 2018; Wan et al., 2019; Wang et al., 2006). Though we could not take into full consideration the sampling days, each set of our samples analyzed in our study are representative and adequate when discussing the abundances and seasonal variations of saccharides as well as when estimating the contributions of biomass burning, plant debris and fungal spores to OC. Each sample set corresponded to specific meteorological condition, which are listed in Table S1.

Compared to the results from previous studies, are there any new findings in Sections 3.1, 3.2 and 3.4? Maybe the authors should focus on the size distributions of saccharides, which may be able to differ the present study from previous ones.

Response:

In Section 3.1, we give a detailed description about the L/M, M/G, L/OC, and L/EC ratios according to particle size. New findings were as followed:

(1) Higher L/M and M/G values were observed in the fine mode, which ascribed to the difference of burning substrates. Hardwood was potentially the burning substrates in the fine mode while mixture impact of hardwood and softwood burning accounted for the relatively lower L/M ratios in the coarse mode. Higher M/G ratios in the fine mode implied the increasing contributions from crop straws burning, especially in haze days.

(2) Dust storms could induce a high L /EC ratio in the coarse mode because of coarse particles brought by dust storms and/or road dust resuspension. While decreased concentrations of EC in the coarse mode in dust storms implying that EC may derived of local emissions rather than long distance transportation of dust.

In Section 3.4, we calculated the contributions of OC from BB, plant debris and fungal spores in terms of particle size. The contributions of BB-OC were different in the fine and coarse mode. BB-OC dominated in the fine mode (>90%) and the contribution of BB-OC in the size range of 2.1–9.0 μm were with a proportion over 60%. While in a larger size range, the relative contribution were below 60% in most periods. Plant debris-derived OC and fungal

spores-derived OC mostly existed in the coarse mode. However, both of them were present in the fine mode, especially in spring and summer. Though the contribution of plant debris and fungal spores in the whole range were insignificant in cold seasons, their relative contribution in the coarse mode were comparatively high.

2、It is completely unclear how the haze, non-haze and dust-storm periods (which were frequently mentioned throughout the manuscript) were identified.

Response:

Thanks for the reviewer for pointing out the missing data. Such information on the meteorological parameters during each sampling period is added in Table S1. The two sets (11–19 April and 4–5 May) were collected in dust storm days, the following sets (30 June–2 July, 12–14 July, 25–27 October, 6–8 November, 27–29 December and 25–27 January) were affected by haze. And the rest sets were for non-haze days.

3、Considering the formation and evolution processes of haze events in Beijing (which could be fast; e.g., Sun et al., Sci. Rep., 2016, 6, 27151), it is questionable whether the so-called haze samples were representative (recalling that only four sets of samples were collected for each season).

Response:

In many previous studies, total suspended particles (TSP) samples were collected by a high-volume sampler, with an operating flow rate of 1.00 $m^3$/min, approximately (Chen et al., 2013; Li et al., 2018; Wan et al., 2019). While in this study, all samples were collected using a nine-stage cascade impactor sampler (Andersen, U.S.A.) at a flow rate of 25.8 L/min from April 2017 to January 2018. Compared to the high-volume sampler, the flow rate of nine-stage cascade impactor sampler is much slower. If the sampling duration was too short, the circumstance of the concentrations of size-resolved samples below the detection line will occur. As a result, we had to prolong the sampling time to guarantee the validity of samples, especially for the non-haze days. Previous studies found that the formation and evolution processes of haze events could be fast, sometimes happened even less in one day (Sun et al., 2014; Yang et al., 2015). To completely encompass the durations of the rapid formation of haze events and the evolution of secondary organic aerosols (which not discussed in this manuscript), we considered 2 to 3 days as a reasonable sampling period. In addition, such sampling period is necessary to collect enough particles for organic analysis.

4、Please clarify why dust storm is a major source of OC in coarse particles. This point was presented as a conclusion but was not clearly explained in the manuscript.

Response:

Thanks for the suggestion. In this manuscript, we presented "Dust storm is a major source of organic compounds in the coarse particles, which induce a remarkable size shift to the coarse mode". This description may be a little inappropriate. We corrected the conclusion in the revised manuscript as "Dust storm greatly enhance organic aerosol concentrations and induce a remarkable size shift towards coarse sizes (see Page 16, Lines 15–16). There are several reasons for this conclusion. Firstly, in general, the concentrations of most primary saccharides sugar alcohols in each impactor stage during dust storms (17–19 April and 4–5 May) were higher than those of non-haze days (19–21 April and 5–6 May), especially for the coarse particle fraction (Figure 6–7). As for anhydrosugars and related sugars, their concentrations in the coarse mode in dust storm were higher, too. Such phenomenon could be probably attributed to strongly windblown mass coarse dust derived from large scale resuspension of dust from crustal, soil, roads or other unpaved areas, along with long-range transport of particles from north and northwest desert regions. Previous studies found that elevated concentrations of trehalose, mannitol and arabitol are generally related to resuspended soil and the outflow of dust storms (Liang et al., 2013; Rogge et al., 2007). Secondly, the GMDs of the total size range and the coarse mode particles in dust storm were larger than non-haze and haze days (Table 4–S5). Some species, such as arabitol, mannitol and inositol, their GMDs associated with coarse particles in dust storm presented a significantly increase, again suggesting the effect of dust storms on the aerosol particle size. Wang et al. (2013) also found that dust storms could act as a major source of coarse particulate matter.

5、A minor point. Page 6, Line 28. Please check the two ratios cited here.

Response:

Thanks. We have corrected the mistake in the revised manuscript (see Page 7, Line 8–9). The revised content is as followed:

"The M/G ratios during all the periods were in a range of 1.35–2.08 with an average 1.70 (Fig. 2b). The M/G ratios maximized in autumn (1.68–1.97, 1.88) and minimized in summer (1.35–1.82, 1.59)."

**Table S1.** Information on the weather conditions during each of the sampling periods from April 2017 to January 2018.

| Year | Season | Sampling period | Duration (min) | T (°C)[a] | RH (%)[b] | WS[c] | PM$_{2.5}$ (μg m$^{-3}$)[d] | PM$_{10}$ (μg m$^{-3}$)[d] | Weather conditions |
|---|---|---|---|---|---|---|---|---|---|
| 2017 | spring | 17–19 Apr. | 2880 | 18.2 | 32.3 | 5 | 174 | 124 | dust storm |
| | | 19–21 Apr. | 2887 | 14.7 | 46.2 | 3 | 78.3 | 84.6 | non-haze |
| | | 4–5 May | 1364 | 20.1 | 33.4 | 6 | 501 | 656 | dust storm |
| | | 5–6 May | 1954 | 17.6 | 25.2 | 5 | 131 | 125 | non-haze |
| | summer | 30 Jun.–2 Jul. | 2862 | 29.3 | 70.5 | 2 | 143 | 103 | haze |
| | | 12–14 Jul. | 2854 | 31.5 | 76.7 | 1 | 89.2 | 75.9 | haze |
| | | 14–16 Jul. | 2900 | 29.0 | 58.3 | 2 | 65.5 | 62.1 | non-haze |
| | | 21–23 Jul. | 2843 | 25.2 | 64.2 | 3 | 42.7 | 36.3 | non-haze |
| | Autumn | 11–13 Oct. | 2877 | 12.3 | 64.2 | 2 | 38.1 | 32.4 | non-haze |
| | | 16–18 Oct. | 2900 | 13.1 | 69.1 | 2 | 78.0 | 68.7 | non-haze |
| | | 25–27 Oct. | 2865 | 11.8 | 81.3 | 1 | 183 | 120 | haze |
| | | 6–8 Nov. | 2887 | 9.36 | 72.4 | 2 | 146 | 91.7 | haze |
| 2017-2018 | Winter | 27–29 Dec. | 2781 | –2.34 | 75.2 | 1 | 137 | 140 | haze |
| | | 2–4 Jan. | 2757 | –2.68 | 32.4 | 3 | 33.7 | 32.1 | non-haze |
| | | 25–27 Jan. | 2858 | –8.13 | 41.6 | 1 | 82.2 | 59.6 | haze |
| | | 29–31 Jan. | 2835 | –1.95 | 22.4 | 2 | 47.5 | 49.5 | non-haze |

[a] temperature (T)
[b] relative humidity (RH)
[c] wind scale (WS)
[d] averaged concentrations from the local monitoring station.

[revised manuscript text omitted]

[a] temperature (T)
[b] relative humidity (RH)
[c] wind scale (WS)
[d] averaged concentrations from the local monitoring station.

**Table S2.** Pearson correlation coefficients (r) among anhydrosugars and carbonaceous components in Beijing aerosols from April 2017 to January 2018.

| | Levoglucosan | Mannosan | Galactosan | OC | EC |
|---|---|---|---|---|---|
| Levoglucosan | 1 | | | | |
| Mannosan | 0.99** | 1 | | | |
| Galactosan | 0.98** | 0.99** | 1 | | |
| OC | 0.16 | 0.21* | 0.20 | 1 | |
| EC | 0.45** | 0.43** | 0.43** | 0.45** | 1 |

*p<0.01.
**p<0.001.

**Table S3.** Pearson correlation coefficients (r) among primary saccharides and sugar alcohols, and with levoglucosan in Beijing aerosols from April 2017 to January 2018.

| | Levoglucosan | Glucose | Fructose | Sucrose | Trehalose | Xylose | Maltose | Arabitol | Mannitol | Inositol | Erythritol | OC | EC |
|---|---|---|---|---|---|---|---|---|---|---|---|---|---|
| Levoglucosan | 1 | | | | | | | | | | | | |
| Glucose | -0.22 | 1 | | | | | | | | | | | |
| Fructose | 0.02 | 0.81** | 1 | | | | | | | | | | |
| Sucrose | -0.03 | 0.81** | 0.74** | 1 | | | | | | | | | |
| Trehalose | -0.30* | 0.54** | 0.48** | 0.30* | 1 | | | | | | | | |
| Xylose | 0.86** | 0.07 | 0.33** | 0.26* | -0.18 | 1 | | | | | | | |
| Maltose | 0.86** | -0.04 | 0.27* | 0.01 | -0.05 | 0.80** | 1 | | | | | | |
| Arabitol | -0.23 | 0.52** | 0.59** | 0.10 | 0.84** | -0.11 | 0.13 | 1 | | | | | |
| Mannitol | -0.28* | 0.59** | 0.02 | 0.66** | -0.19 | 0.02 | 0.83** | 1 | | | | | |
| Inositol | 0.10 | 0.63** | 0.71** | 0.78** | 0.65** | 0.29* | 0.22 | 0.36** | 0.12 | 1 | | | |
| Erythritol | 0.50** | 0.01 | 0.24 | 0.27* | 0.03 | 0.40** | 0.59** | 0.42** | 0.34** | -0.16 | 1 | | |
| OC | 0.16 | 0.21 | 0.26* | 0.24 | 0.82** | 0.25 | 0.40** | 0.77** | 0.49** | 0.75** | 0.33** | 1 | |
| EC | 0.45** | 0.32** | 0.59** | 0.24 | 0.16 | 0.62** | 0.69** | 0.45** | 0.35** | 0.24 | 0.65** | 0.45** | 1 |

*p<0.01.
**p<0.001.

**Table S4.** Geometric mean diameter (GMD, μm) of anhydrosugars and other saccharides in the fine mode particles (<2.1 μm) in the Beijing, from April 2017 to January 2018. GMD: $logGMD = (\sum C_i logD_{pi})/\sum C_i$, where $C_i$ is the concentration of compound in size i and $D_{pi}$ is the geometric mean particle diameter collected on stage i.

| Compounds | Fine | | | | | | | |
| --- | --- | --- | --- | --- | --- | --- | --- | --- |
| | Spring | | Summer | | Autumn | | Winter | |
| | dust storm | non-haze | haze | non-haze | haze | non-haze | haze | non-haze |
| *Anhydrosugars* | | | | | | | | |
| Galactosan | 0.51 | 0.46 | 0.48 | 0.46 | 0.76 | 0.67 | 0.61 | 0.47 |
| Mannosan | 0.55 | 0.46 | 0.62 | 0.55 | 0.74 | 0.67 | 0.61 | 0.44 |
| Levoglucosan | 0.57 | 0.41 | 0.59 | 0.56 | 0.72 | 0.50 | 0.55 | 0.45 |
| *Sugar Alcohols* | | | | | | | | |
| Arabitol | 0.38 | 0.43 | 0.69 | 0.84 | 0.58 | 0.75 | 0.61 | 0.37 |
| Mannitol | 0.46 | 0.42 | 0.76 | 0.49 | 0.69 | 0.55 | 0.53 | 0.54 |
| Inositol | 0.34 | 0.40 | 0.54 | 0.39 | 0.51 | 0.46 | 0.50 | 0.37 |
| Erythritol | 0.39 | 0.47 | 0.51 | 0.60 | 0.71 | 0.55 | 0.61 | 0.51 |
| *Sugars* | | | | | | | | |
| Glucose | 0.38 | 0.46 | 0.72 | 0.68 | 0.60 | 0.48 | 0.51 | 0.43 |
| Sucrose | 0.33 | 0.51 | 0.68 | 0.48 | 0.65 | 0.56 | 0.48 | 0.49 |
| Fructose | 0.40 | 0.44 | 0.46 | 0.49 | 0.65 | 0.43 | 0.48 | 0.36 |
| Maltose | 0.31 | 0.36 | 0.47 | 0.31 | 0.57 | 0.39 | 0.61 | 0.34 |
| Xylose | 0.43 | 0.42 | 0.62 | 0.66 | 0.64 | 0.65 | 0.66 | 0.41 |
| Trehalose | 1.02 | 0.69 | 0.87 | 0.64 | 0.80 | 0.62 | 0.70 | 0.55 |

**Table S5.** Geometric mean diameter (GMD, μm) of anhydrosugars and other saccharides in the coarse mode particles (>2.1 μm) in the Beijing, from April 2017 to January 2018.

| Compounds | Coarse | | | | | | | |
|---|---|---|---|---|---|---|---|---|
| | Spring | | Summer | | Autumn | | Winter | |
| | dust storm | non-haze | haze | non-haze | haze | non-haze | haze | non-haze |
| *Anhydrosugars* | | | | | | | | |
| Galactosan | 5.10 | 5.04 | 5.39 | 5.97 | 4.55 | 4.51 | 4.25 | 5.32 |
| Mannosan | 4.79 | 4.85 | 5.40 | 5.56 | 3.90 | 4.29 | 3.76 | 6.01 |
| Levoglucosan | 4.54 | 4.47 | 5.18 | 5.08 | 4.00 | 4.04 | 3.74 | 4.44 |
| *Sugar Alcohols* | | | | | | | | |
| Arabitol | 11.4 | 9.14 | 5.33 | 4.34 | 5.35 | 5.21 | 6.10 | 6.80 |
| Mannitol | 10.7 | 8.71 | 5.38 | 4.68 | 4.87 | 5.91 | 5.81 | 7.04 |
| Inositol | 11.47 | 8.67 | 5.37 | 5.51 | 5.42 | 5.59 | 6.52 | 7.39 |
| Erythritol | 6.09 | 5.82 | 5.89 | 6.35 | 5.27 | 5.46 | 5.97 | 6.23 |
| *Sugars* | | | | | | | | |
| Glucose | 10.6 | 8.69 | 5.00 | 4.43 | 5.79 | 5.28 | 6.88 | 7.43 |
| Sucrose | 7.03 | 9.01 | 6.21 | 6.31 | 5.65 | 5.38 | 7.16 | 7.55 |
| Fructose | 8.24 | 3.87 | 5.80 | 5.04 | 6.15 | 4.42 | 7.29 | 7.32 |
| Maltose | 7.13 | 6.18 | 5.92 | 5.55 | 5.33 | 5.83 | 6.31 | 6.43 |
| Xylose | 6.35 | 6.27 | 5.47 | 4.78 | 5.20 | 5.38 | 5.25 | 5.61 |
| Trehalose | 7.41 | 7.00 | 5.64 | 4.97 | 5.39 | 5.58 | 6.86 | 7.91 |

[Figure]

**Figure S1.** Clusters of 3-day backward trajectories of air masses arriving at Beijing (500 m a.g.l.) during dust storm events and haze periods from April 2017to January 2018. The numbers in each panel imply the percentages of trajectories in the sampling periods with air mass origins.